# Review on Compounds Isolated from Eriocaulaceae Family and Evaluation of Biological Activities by Machine Learning

**DOI:** 10.3390/molecules27217186

**Published:** 2022-10-24

**Authors:** Laysa Lanes Pereira Ferreira Moreira, Renata Priscila Barros de Menezes, Luciana Scotti, Marcus Tullius Scotti, Valdemar Lacerda Júnior, Warley de Souza Borges

**Affiliations:** 1Post-Graduate Program in Chemistry, Federal University of Espírito Santo, Av. Fernando Ferrari, 514, Vitória 29075-910, Brazil; 2Post-Graduate Program in Natural Synthetic Bioactive Products, Federal University of Paraiba, R. Table Stanislau Eloy, 41—Conj. Pres. Castelo Branco III, João Pessoa 58050-585, Brazil

**Keywords:** Eriocaulaceae, multitarget potential, ligand-based virtual screening, flavonoids, naphthopyranones, xanthones

## Abstract

Eriocaulaceae is a pantropical family whose main center of biodiversity is in Brazil. In general, the family has about 1200 species, in which phytochemical and biological studies have shown a variety of structures and activities. The aim of this research is to compile the compounds isolated in the Eriocaulaceae family and carry out a computational study on their biological targets. The bibliographic research was carried out on six databases. Tables were built and organized according to the chemical class. In addition, a summary of the methods of isolating the compounds was also made. In the computational study were used ChEMBL platform, DRAGON 7.0, and the KNIME 4.4.0 software. Two hundred and twenty-two different compounds have been isolated in sixty-eight species, divided mainly into flavonoids and naphthopyranones, and minor compounds. The ligand-based virtual screening found promising molecules and molecules with multitarget potential, such as xanthones **194**, **196**, **200** and saponin **202**, with xanthone **194** as the most promising. Several compounds with biological activities were isolated in the family, but the chemical profiles of many species are still unknown. The selected structures are a starting point for further studies to develop new antiparasitic and antiviral compounds based on natural products.

## 1. Introduction

Eriocaulaceae is a pantropical family, which comprises around 1200 species divided in ten genera [1]. It is easily distinguished from other families due to most of its species presenting short stems, leaves in a rosette form, and long scapes grouped in capitula [2,3].

Brazil is the main center of Eriocaulaceae biodiversity, where there are a total of 610 species, 548 of them being endemic, and distributed in eight genera, with one endemic genus. Some species, belonging to *Syngonanthus* and *Comanthera* genera, are economically important, being commercialized as ornamental objects and commonly exported as “everlasting plants.” In addition to those, in regions called “campos rupestres,” in the states of Bahia, Goiás, Minas Gerais, and Tocantins, the plants that occur naturally are one of the main sources of income for local inhabitants [3,4,5].

Among the genera from Eriocaulaceae, *Eriocaulon* is the biggest, with 478 species, and the only one with a pantropical distribution. It is an aquatic and wetland genus. In Brazil it occurs in all domains of the country and has 61 known species. The greatest diversity of *Eriocaulon* is in the Cerrado, where about 80% of species are endemic [6,7,8].

*Paepalanthus* is the second most expressive genus in number of species, comprising around 400 species and occurring specially in the Americas and a few species in Africa [7,9]. In Brazil, 348 species of *Paepalanthus* can be found, thus making it the third largest genus in Brazilian flora and the largest among Brazilian monocots [9,10,11]. Despite the many species, phytochemical studies on the genus have focused on only 16 species. 

Natural products are a great source of active principles, and many molecules have potential activity for various biological functions [12]. A study by Newman and Gragg (2020) found that 68.8% of new accepted drugs are either natural products or inspired by natural product molecules [12].

Thus, natural products are widely used in the virtual screening technique, which seeks by computational tools, in silico, to identify molecules with active potential [13,14]. The ligand-based virtual screening is a computational technique that uses pools of molecules with biological activity information to build predictive models and then predict activity pools of molecules whose activity information for the selected biological functions is unknown [14,15,16]. It is a very effective tool in the search for new drugs and avoids unnecessary laboratory experiments, thus reducing the analysis cost and time [14,15,16].

Neglected diseases are a group of diseases common among the world’s poorest populations, which suffer from lack of resources, poor health infrastructure, and substandard sanitation [17,18,19]. They are a global health problem. The World Health Organization classifies 17 diseases as neglected diseases, including leishmaniasis, Chagas disease, schistosomiasis, dengue fever, etc. [20,21].

Dokkedal et al. [22] published the first and only article on compounds isolated from the Eriocaulaceae family in 2008. The authors described flavonoids, naphthopyranones, and xanthones from *Paepalanthus*, *Syngonanthus*, and *Leiothrix*. Thus, this study aimed to carry out a bibliographic search on compounds isolated from the family from the first paper to the current year and to evaluate their possible biological activities against neglected diseases by ligand-based virtual screening. This research found other compounds from the mentioned classes plus saponins, quinones, and others. The computational study evaluating neglected diseases showed saponins and xanthones had the best results.

## 2. Results

### 2.1. Chemical Constituents of Eriocaulaceae 

Table 1, Table 2, Table 3, Table 4, Table 5, Table 6, Table 7 and Table 8 show the structures of the compounds **1–222**, isolated from sixty-eight species, from different organs and fungi. The most predominant metabolites are flavonoids and naphthopyranones, which have been used as chemical markers for some taxa in Eriocaulaceae, mainly *Paepalanthus* genus [23,24]. Other classes such as xanthones, saponins, steroids, anthraquinones, naphthoquinones, and minor derivatives such as tocopherol, spirocyclic lactams, curvulinic acid, and caffeic acid were also isolated.

#### 2.1.1. Flavonoids

Flavonoids are a class of aromatic and polyphenolic compounds that come in various structural forms and are divided into groups according to the oxidation and unsaturation of the C-ring [25]. Those compounds have several biological activities such as antioxidant, antitumoral, anti-inflammatory, antiviral, and antimicrobial, among others [25,26,27,28].

The first paper found about flavonoids compounds in the Eriocaulaceae family indicated their isolation from *Eriocaulon* genus. Six species were studied by Bate-Smith and Harborne (1969), where quercetagetin (**1**) was isolated from *Eriocaulon septangulare* and *E. brownianum*. According to the authors, compound (**1**) was previously reported in *E. nilagirense*, *E. decangulare*, *E. sexangulare*, and *E. wightianum*. Compound (**2**) was isolated from *E. brownianum* and *E. truncate*, in a crystalline form, showing a bright yellow-brown color in UV light. That compound was identified as patuletin, the 6-methyl ether of quercetagetin. Quercetin (**3**) was also found in *E. brownianum* [29]. According to Dokkedal et al. (2008), (**1**) was reported in *Paepalanthus ramosus* and (**2**) in *P. macrocephalus* and *P. macropodus* [22].



Patuletin (**2**) was also isolated from *Eriocaulon buergerianum*. The whole plant was extracted with 95% ethanol and the concentrated crude extract was partitioned with EtOAc. The EtOAc extract was submitted to separation by chromatography in a silica gel column. Compound (**2**) had effects on the gene expression and activity of fatty acid synthase (FASN) in the human breast cancer SK-BR-3 cell line, and apoptotic effects on breast cancer cells. MTT assays and flow cytometry were also used to measure cell growth and cell apoptosis, respectively. As a result, patuletin may be considered a natural inhibitor of FASN, inducing anti-proliferative and pro-apoptotic effects in human breast cancer [30].

Dokkedal and Salatino (1992), studying *Leiothrix* species, extracted flavonoids from dried and powdered leaves with 80% MeOH. In addition to Luteolin C-glycoside, the authors also isolated compounds **4**, **5**, **6**, **7**, **8**, and **9**, which were identified by chromatographic techniques and compared with known compounds. All compounds are flavone derivatives, based on luteolin structure [31].



A study by Mayworm and Salatino (1993) isolated flavonoids **2**, **10**–**15** from capitula of *Paepalanthus* species. Methanolic extracts of *Paepalanthus bifrons*, *P. hilairei*, *P. planifolius*, and *P. robustus* were submitted to preparative chromatography and the flavonoids were purified in a Sephadex LH-20 column [32].



Flavonoids were quantified in methanolic aerial part extracts of *Paepalanthus giganteus* and *Syngonanthus nitens* by LC-PDA with co-injection experiments. Quercetin (**3**) was identified in capitula and scapes of *P. giganteus* and luteolin (**12**) in capitula and scapes of *S. nitens* [33].

Twenty-two species from *Syngonanthus* genus were phytochemically studied. From them, thirty-nine flavone glycosides have been identified, but not all could be drawn due to lack of information. The dried and powdered leaves of *Syngonanthus* spp. were extracted with 80% MeOH under reflux. The fractionation was performed with preparative chromatography and/or column chromatography. Flavonoids mono, di, and triglycosides (**4**, **16–26**) were identified by thin layer chromatography (TLC) and glycosides by standard methods [34].



Studying EtOH extracts of the capitula of *Paepalanthus polyanthus*, *P. hilairei*, *P. robustus*, *P. ramosus*, and *P. denudatus*, [22] isolated two new and taxonomically relevant acylated flavonoids, 6-methoxykaempferol-3-*O*-β-D-6′’-(*p*-coumaroyl)-glucopyranoside (**27**) and 6-methoxyquercetin-3-*O*-β-D-6′’-(*p*-coumaroyl)-glucopyranoside (**28**). In addition, flavonoids quercetagetin (**1**), patuletin (**2**), quercetagetin-7-*O*-glucopyranoside (**29**), patuletin-3-*O*-glucopyranoside (**30**), patuletin-3-*O*-rutinoside (**31**), 6-methoxykaempferol (**32**), and 6-methoxykaempferol-3-*O*-glucopyranoside (**33**) were also isolated. Their structures were determined by spectroscopic and spectrometric methods [35]. Additionally, according to Dokkedal et al. (2008), **30** and **32** were reported in *P. brachypu*, **32** and **33** in *P. vellozioides* and *P. latipes*, and **33** in *P. planifolius* [22].



One aglycone (**34**) and five glycosylated flavonoids (**35–39**) were obtained from fresh leaves of *Paepalanthus vellozioides* and *P. latipes* after extraction with ethanol and chromatographic procedures. Most compounds have the glycoside group attached to the 3-position of the flavonoid skeleton [24]. Additionally, according to Dokkedal et al. (2008), **34** and **36** were isolated in *P. bromelioides* and **35** in *P. planifolius* [22].



Moreira and collaborators (2002) studied the ethanol extract of scapes of the *Paepalanthus latipes*. Two flavonoids (**34** and **36**) were isolated, and the mutagenic activity of the metabolites and the crude extract was evaluated against different mutant strains of *Salmonella typhimurium* (TA98 and TA100). None of the flavonoids nor the crude extract showed mutagenic activity induction in bacteria [36]. Eleven years later, another study with aerial parts of the same plant obtained the same metabolites. Antimycobacterial activity was performed against *Mycobacterium tuberculosis* (H37Rv ATCC 27294) and *Mycobacterium avium* (ATCC 15769) and showed a low activity of the flavonoids and ethanolic extract [37].

Santos et al. (1999) obtained ethanol extracts from four different species of *Paepalanthus*: *P. hilairei*, *P. bromelioides*, *P. vellozioides*, and *P. latipes*. Nine flavonoid metabolites (**32**, **33**, **2**, **40**, **29**, **41**, and **42**) were obtained after fractionation by droplet countercurrent chromatography (DCCC) followed by column chromatography [38]. Dokkedal et al. (2008) also reported the presence of **32** in *P. macropodus* and **41** in *Leiothrix curvifolia* in their review [22].



The on-line separation and identification of two 6-methoxykaempferol glycosides previously isolated (**28** and **33**) has been performed with an HPLC-NMR coupling using C-30 phase. Flavonoids were isolated from *Paepalanthus ramosus* capitula ethanolic extracts [39].

In 2001, studying the phytochemistry of *Leiothrix curvifolia* and *L. flavescens*, Santos and collaborators isolated seven flavonoids from EtOH extracts after separation in Sephadex LH-20. The flavonoids **10**, **12**, and **43** were isolated in *L. curvifolia* and the flavonoids **44–48** in *L. flavescens* [40]. According to Dokkedal et al. (2008) in their review, compound **47** was also found in *Syngonanthus bisulcatus* [22].



Dried leaves of *Paepalanthus chlorocephalus* and *Paepalanthus argenteus* var. *argenteus* were used to prepare a methanolic extract. The analysis of the extracts resulted in the identification of the constituents jaceosidin (**41**) and its glycosylated derivative, jaceosidin-7-glycoside (**46**), for the extract of *P. argenteus* var. *argenteus*, whereas for *P. chlorocephalus* the metabolite nepetin (6-methoxyluteolin, **7**) was identified [41].

Capitula and scapes of *Paepalanthus polyanthus* were powdered and extracted with different organic solvents. The EtOH extract was chromatographed on a Sephadex LH-20 column with MeOH as eluent. This procedure led to isolation of five flavonoids: 6-methoxyquercetin-3-*O*-β-D-glucopyranoside (**30**), patuletin-3-*O*-β-D-rutinoside (**31**), quercetagetin 7-methyl ether-3-*O*-neohesperidoside (**38**), 5,6,7,8,3′,4′-hexahydroxyflavonol (**49**), and the new 6-methoxyquercetin-3-*O*-(6′’-*E*-feruloyl)-β-D-glucopyranoside (**50**) [42].



Capitula of *Paepalanthus macropodus* were powdered and extracted successively with different solvents. The EtOH extract was chromatographed on a Sephadex LH-20 column with MeOH as eluent. Three flavonoids metabolites were isolated: 6-methoxyquercetin-3-*O*-β-D-glucopyranoside (**30**), 5,6,7,8,3,4-hexahydroxyflavonol (**49**), and quercetagetin-6-methyl-ether-3-*O*-β-D-glucopyranosyl(1→4)-glucopyranoside (**51**) [43].



Four flavonoids, (2*S*)-3′,4′-methylenedioxy-5,7-dimethoxyflavan (**52**) and hispidulin 7-(6-*E*-*p*-coumaroyl-β-D-glucopyranoside) (**53**), hispidulin (**54**), and hispidulin-7-*O*-glucoside (**55**), were isolated from the capitula of *Eriocaulon buergerianum* Koern. Their structures were established by spectral and chemical evidence [44].



The ethyl acetate fraction of *Leiothrix flavescens* was fractionated by high-speed counter-current chromatography (HSCCC) using the mixture of *n*-hexane:ethyl acetate:methanol:water, 1:5:1:5 (*v*/*v*/*v*/*v*). The lower phase of the solvent system was used as a stationary phase and the upper one as a mobile phase. Some fractions provided yellow solids identified as flavonoids when sprayed with NP/PEG reagent. The substances were purified in a Sephadex LH-20 column and eluted with methanol. This procedure identified apigenin (**10**) luteolin (**12**), and 6-methoxyluteolin (**54**). The antioxidant activity of methanol extract was evaluated by the DPPH method, showing that the crude extract is more active at a concentration above 80.0 g mL^−1^ when compared with the acid gallic standard. Phenolic compounds were also quantified by using the Folin–Ciocalteau reagent method. According to the authors, the total phenol concentration in 1.0 g of the methanol extract is 47.0 mg [45].

Three flavonoids, 3′,4′,5,6,7,8-hexahydroxyflavone (**48**), 3′,4′,5,6,7-pentahydroxyflavone (**56**), and 3′,4′,5,6-tetrahydroxy-7-*O*-β-D-glucopyranosilflavone (**57**), were obtained from ethanolic extract of *Paepalanthus microphyllus* capitula [46]. According to Dokkedal et al. (2008) in their review on *Paepalanthus* chemistry, compound **56** was also isolated in *Paepalanthus robustus*, *P. planifolius*, and *P. flavescens* and **57** was also found in *P. planifolius* and *P. flavescens* [22].



Capitula of *Eriocaulon ligulatum* were powdered and macerated with *n*-hexane, methylene chloride, and MeOH. Methanolic extract was submitted to Sephadex LH-20 column, followed by separation in HPLC using refractive detector. This procedure gave rise to two new (**58**, **59**) and six known (**7**, **8**, **54**, **55**, **60, 61**) flavonoids which were identified by spectroscopic and spectrometric methods [47]. Additionally, 6-methoxyluteolin-7-*O*-β-D-allopyranoside (**58**) and 6-methoxyapigenin-7-*O*-β-D-allopyranoside (**59**) were isolated from the same part and species by HSCCC using a mixture of two phases composed of ethyl acetate:*n*-propanol:water (140:8:80), *v*/*v*/*v*. This separation was performed in 3 h and the two flavonoids were identified by NMR and ESI-MS, comparing their spectra with published data [48].



Aerial parts of *Syngonanthus bisulcatus* were extracted with ethanol and submitted to separation with Sephadex LH-20 and eluted with methanol. Some fractions were purified by different methods of chromatography to separate the flavonoids luteolin (**12**), 5,6,3′,4′-tetrahydroxy-7-*O*-β-D-glucopyrade (**57**), 5-hydroxy-7,4′-dimethoxy-6-*C*-β-D-glucopyranosylflavone (**62**), isovitexin (5,7,4′-trihydroxy-6-*C*-β-D-glucopyranosylflavone, **63**), and lutonarin (5,3′,4′-trihydroxy-6-*C*-7-O-β-D-glucopyranosylflavone, **64**). The structure of the compounds was characterized by spectroscopic and spectrometric methods. The ethanol extract was also examined in the ulcer model ethanol/HCl-induced gastric mucosal lesions, showing a significant inhibition of ulcer formation when compared with the control group [49].



The work carried out by Dokkedal et al. (2007) resulted in the isolation of a dihydroflavonol C-glycoside characterized as xeractinol (**65**) from a methanol extract of the leaves of *Paepalanthus argenteus* var. *argenteus*. Xeractinol was isolated after eluting from a Sephadex LH-20 column with MeOH and showed a yellow spot on TLC under UV light [50].



The *n*-butanol fraction from *Eriocaulon ligulatum* capitula was fractionated on a HSCCC apparatus with four gradients. This procedure gave two known flavonoids, 6-methoxyapigenin-7-*O*-β-D-glucopyranoside (**55**) and 6-methoxyapigenin-7-*O*-β-D-allopyranoside (**59**), and the new acylated flavonoid, 6,4′-dimethoxyquercetin-3-*O*-β-D-6″[3″′,4″′,5″′-trihydroxy-(*E*)-cinnamoyl]-glucopyranoside (**66**). Compounds were characterized by spectroscopic and spectrometric methods. The mutagenic effect of the *n*-BuOH extract was evaluated and showed mutagenic activity in the *Salmonella*/microsome assay, in TA100, TA97a, and TA102 strains, and for dichloromethane extract tested in TA98 strain [51].



Bosqueiro (2000), studying the chemistry of *Paepalanthus* spp. from different taxa, observed that chemical profiles are very distinct. Three flavonoids, **2**, **7**, and **32**, were isolated in *Paepalanthus chlorocephalus*. Flavonoids **2**, **41**, and **4** were isolated in *P. argenteus*; **28**, **30**, **32**, and **69** in *P. macrocephalus*; **34–36**, **40**, **67**, and **68** in *P. vellozioides* and *P. latipes*; **27**, **29**, **30**, **32**, and **33** in *P. denudatus* [22,52]. According to Dokkedal et al. (2008), 6,3′-dimethoxyquercetin-7-*O*-β-D-glucopyranoside (**69**) was also isolated in *P. bromelioides*, and apigenin-6-*C*-8-*C*-glucopyranoside (**70**) was also isolated in *P. planifolius* [22].



The whole plant of *Eriocaulon buergerianum* was extracted with 95% EtOH and fractionated with petroleum ether, EtOAc, and *n*-BuOH. All extracts were purified using different chromatographic methods. The EtOAc extract contained the compounds **2**, **30**, **46**, **50**, **54**, **55**, **71**, **72**, and **75** and *n*-BuOH extract contained **40**, **50**, and **73**. Antibacterial assays were performed to all isolated compounds, using the standard *Staphylococcus aureus* strain (ATCC 25923). As a result, ten compounds exhibited antibacterial activity with minimum inhibitory concentrations (MICs) ranging from 32 to 256 µg mL^−1^ [53]. Additionally, patuletin (**2**) and hispidulin (**54**) were evaluated for inhibitory activities on fatty acid synthase, showing significant activity [54].



Methanol extract of *Eriocaulon ligulatum* capitula was submitted to liquid chromatography-electrospray ionization multistage ion trap mass spectrometry (LC-ESI-IT-MS^n^). As a result, the authors identified **3**, **10**, **12**, **54**, **59**, and **76–81** based on their fragmentation patterns in MS experiments and on NMR analysis for isolated compounds [55].



The compounds 6-methoxyapigenin (**10**), 6-methoxyapigenin-7-*O*-β-D-glucopyranoside (**17**), and rutin (**82**) were quantified by using high performance liquid chromatography with DAD detection in methanolic extracts of capitula of *Syngonanthus suberosus*, *S. dealbatus*, *Eriocaulon ligulatum*, and capitula and leaves *of Leiothix spiralis*, in time intervals of less than 7 min. The identification was made by comparing the retention times with those of standards, by adding standard solutions to the samples analyzed by HPLC, and by comparing their UV-Vis spectrum. All extracts were tested against six strains of microorganism, inhibiting the growth of all the tested microorganisms [56].



An analytical study was carried out with scapes and flowers of *Syngonanthus nintens* to define the metabolite fingerprint by HPLC-ESI-MS^n^. Additionally, the methanolic extracted of both scapes and flowers were filtered and fractionated on a Sephadex LH-20 column, using MeOH as mobile phase. Other separation procedures were performed to isolate **12**, **45**, **47**, **56**, and **83–90** from scapes. The structures of all compounds were elucidated by NMR spectroscopic data [57]. The in vitro antioxidant properties of the *S. nintens* methanolic extract were evaluated by electron paramagnetic resonance (EPR) spectroscopy based on their ability to scavenge the DPPH radical. The kinetics of reaction between DPPH and *S. nintens* was determined. Luteolin (**12**) and isoorientin (**47**) were also used to investigate kinetics of reaction between DPPH and flavonoids. As a result*, S. nitens* showed a high antioxidant capacity the authors attributed to the presence of flavonoids, and **47** presented an antioxidant activity 40% higher than **12** [58].



An ethanolic extract from scapes of *Syngonanthus macrolepis* yielded a flavonoid-rich fraction after going through a Sephadex LH-20 chromatographic column, containing luteolin (**12**), 7-methoxyluteolin-6-*C*-β-D-glucopyranoside (**45**), luteolin-6-*C*-β-D-glucopyranoside (**47**), 6-hydroxyluteolin (**56**), and 7,3′-dimethoxyluteolin-6-*C*-β-D-glucopyranoside (**88**) identified after HPLC purification. A flavonoid-rich fraction was investigated for preventing gastric ulceration in mice and rats, showing a significant reduction in gastric injury in all models tested, without altering gastric juice parameters after pylorus ligation [59].

A methanolic extract of *Leiothrix spiralis* leaves was chromatographed on a Sephadex LH-20 column with MeOH as eluent. After separation of three fractions by medium pressure liquid chromatography (MPLC), five flavonoids (**44**, **45**, **47**, **87**, and **91**) were found. Minimum inhibitory concentration (MIC) analysis revealed antibacterial and/or antifungal activity for all compounds [60].



A study carried out with *Paepalanthus geniculatus* Kunth. flowers resulted in several metabolites of different classes, identifying 16 flavonoids. Most of them were derived from quercetagetin (**67**, **96–98**) and galetine (**92–94**, **100**). Other metabolites such as **28**, **30**, **33**, **67**, **99**, and **100** and a truxilate metabolite (**101**), which is a glucoside cyclodimer, were also isolated. Furthermore, **31**, **95**, and **97** showed significant antioxidant activity [61]. Previous and subsequent works reported the isolation of 6-methoxykaempferol-3-*O*-glucopyranoside (**33**) in capitula of *Paepalanthus bromeloides* and *P. macropodus* [38,43].



A methanolic extract of *Eriocaulon australe* R. Br. capitula was partitioned with EtOAc and submitted to silica gel column chromatography using increasing MeOH in CHCl_3_ (0–40%, *v*/*v*) to give four fractions. Those fractions were submitted to different types of separation resulting in the compounds **17**, **41**, **46**, **53–55**, and **102–109**. The in vitro cytotoxicity of the compounds was evaluated using the MTT colorimetric assay. Compounds **41**, **54**, and **102** were cytotoxic to A549; **41**, **54**, **102–104**, and **106** to MCF-7; and **54**, **41**, and **103** to HeLa cells [62].



Methanolic extracts of capitula and scapes of *Syngonanthus dealbatus*, *S. macrolepsis*, *S. nitens*, and *S. suberosus* were separated with Sephadex LH-20. Fractions of scapes column were also separated in HSCCC. Compounds **45** and **87** were purified by HPLC-UV and **12**, **56**, and **88** by HPLC-RID. The mutagenicity of extracts and isolated compounds were evaluated, but none of them showed activity. All isolated flavones could also be used as new antimutagenic agents [63].

Methanolic extracts from powdered capitula and scapes of *Paepalanthus chiquitensis* Herzog were fractioned in Sephadex LH-20 using MeOH as eluent. A fraction from a mentioned procedure of capitula yielded the pure compound **28**. A fraction from scapes was separated by semi-preparative HPLC-IR yielding the new compound, **113**. Compound **120** was also isolated from both capitula and scapes. Other compounds such as **28**, **76**, and **114** were identified by HPLC-ESI-MS^n^ using external standard and compounds **80**, **110–112**, and **115–119** according to *m/z* and literature. A *Salmonella*/microsome biological assay was performed with methanolic extracts, resulting in mutagenic activity against the TA97a strain [64].



The hydroethanolic extract of the aerial parts from *Tonina fluviatilis* was submitted to different methods of separation to yield 6-methoxyquercetin-3-*O*-β-D-glucopyranoside (**30**), 6-hydroxy-7-methoxyquercetin-3-*O*-β-D-glucopyranoside (**35**), and 6,7-dimethoxyquercetin-3-*O*-β-D-glucopyranoside (**121**). After obtaining the compounds and elucidating their structures, they were quantified in the extracts by using HPLC-DAD. The radical scavenging activity was also performed to extract and isolate compounds, showing a better result for compounds **30** and **35** [1].



Methanolic extract of the *Paepalanthus acanthophyllus* capitula was fractionated by HPLC-PDA semipreparative, allowing the isolation of 6-methoxykaempferol-3-*O*-β-D-glucopyranoside (**33**), 6-methoxykaempferol-3-*O*-(6”-*p*-coumaroyl)-β-D-glucopyranosyl-7-*O*-β-D-glucopyranoside (**122**), and 6-methoxykaempferol-3-7-di-*O*-β-D-glucopyranoside (**123**) [65].



The hydroethanolic extract of *Eriocaulon buergerianum* Koern. was partitioned with EtOAc, BuOH, and H_2_O. After repeated and different chromatographic methods, the EtOAc fraction yielded compounds **10**, **41**, and **54**; the BuOH fraction compounds **17**, **46**, **53**, **55**, and **124**; and the H_2_O fraction compounds **53** and **124** [66].



Qiao et al. (2012), studying the chemical constituents of a Chinese herbal medicine Gu-Jing-Cao (*Eriocaulon buergerianum*) to identify—by high-performance liquid chromatography with diode array detection and electrospray ionization tandem mass spectrometry (HPLC-DAD-ESI-MS^n^)—adulterating species, identified, in *Eriocaulon sexangulare*, the flavonoids **30**, **54**, **55**, **104**, **126**, and **129**; in *E. buergerianum*, the flavonoids **2**, **30**, **31**, **54**, **55**, **71**, **73**, **104**, **105**, **125**, and **129**; in *E. cinereum*, the flavonoids **2**, **30**, **31**, **54**, **55**, **71**, **73**, **104**, **105**, **111**, **125**, **127**, **129**, and **130**; in *E. faberi*, the flavonoids **2**, **30**, **31**, **54**, **55**, **71**, **73**, **104**, **105**, **125**, and **130**. The authors did not show in what species compound **128** was isolated. Besides the mentioned compounds, other compounds not completely elucidated were found [67].



In summary, one hundred and thirty compounds were isolated from one species of *Tonina*, six species of *Leiothrix*, twelve species of *Eriocaulon*, nineteen species of *Syngonanthus,* and twenty species of *Paepalanthus*, being thirty-two aglycones and ninety-eight glycones.

Several subclasses of flavonoids have been isolated, such as flavones, flavonols, flavanols, flavanonols, flavanone, and isoflavones, more specifically, derivatives of apigenin, luteolin, quercetin, kaempferol, genistein, and others.

Regarding substituents, 6-methoxylated flavonoids are very common (45.9%), but the substitution at the positions C-3 (2.4%), C-5 (4.1%), C-7 (19.7%), C-3′ (11.5%), C-4′ (10.7%), and C-5′ (5.7%) also occur. Besides this, in this study, we can observe the presence of six specific kinds of sugar, such as glucose (Glu), rhamnose (Rha), xylose (Xyl), allose, arabinose (Ara), galactose (Gal), and its combinations, forming disaccharides. However, the glycones with glucose are the most common, especially at C-3 and C-7 positions, but also found at C-6, C-8, and C-4′. Glucoside substituents such as caffeoyl, coumaroyl, feruloyl, and acetyl were also found.

Table 1 shows the names of flavonoids, the part of the plant where they were isolated, species, and authors.

**Table 1 molecules-27-07186-t001:** Flavonoids isolated from Eriocaulaceae species.

Compounds	MF	[M-H]^−^	Organ	Species	Reference
Quercetagetin **(1)**	C_15_H_10_O_8_	317	Leaves	*Eriocaulon septangulare*	[29]
			Leaves	*E. brownianum*	[29]
			-	*E. nilagirense*	[29]
			Leaves	*E. decangulare*	[29]
			Leaves	*E. sexangulare*	[29]
			Leaves	*E. wightianum*	[29]
			Capitula	*Paepalanthus polyanthus* *P. robustus* *P. ramosus*	[35]
			-	*P. ramosus*	[22]
Patuletin **(2)**	C_16_H_12_O_8_	331	Leaves	*E. brownianum*	[29]
			Leaves	*E. truncatum*	[29]
			Capitula	*P. planifolius*	[32]
			Capitula	*P. polyanthus*	[35]
			Capitula	*P. bromelioides* *P. latipes*	[38]
			Capitula	*P. chlorocephalus* *P. argenteus* var. *argenteus*	[52]
			Capitula	*P. macrocephalus*	[22]
			whole plant	*E. buergerianum*	[53]
			whole plant	*E. buergerianum*	[30]
			Capitula/ seeds	*E. buergerianum* *E. cinereum* *E. faberi*	[67]
			-	*P. macropodus*	[22]
Quercetin **(3)**	C_15_H_10_O_7_	301	Leaves	*E. brownianum*	[29]
			Capitula	*E. ligulatum*	[55]
			Aerial part	*P. giganteus*	[33]
Luteolin 7-*O*-glucoside **(4)**	C_21_H_20_O_11_	447	Leaves	*Leiothrix curvifolia var*. *mucronata*	[31]
Luteolin 7-*O*-triglucoside **(5)**	-	-	Leaves	*L. curvifolia* *L. curvifolia var. mucronata* *L. plantago* *L. sclerophylla* *L. vivípara*	[31]
Luteolin 7-*O*-diarabinoside **(6)**	-	-	Leaves	*L. spiralis*	[31]
Nepetin **(7)**	C_16_H_12_O_7_	315	Leaves	*L. curvifolia var. curvifolia*	[31]
			Capitula	*P. chlorocephalus*	[52]
			Leaves	*P. chlorocephalus*	[41]
			Capitula	*E. ligulatum*	[47]
Nepetin 7-*O*-glucoside **(8)**	C_22_H_22_O_12_	477	Leaves	*L. curvifolia var*. *mucronata*	[31]
			Capitula	*E. ligulatum*	[47]
Nepetin 7-*O*-arabinoside **(9)**	C_21_H_20_O_11_	447	Leaves	*L. curvifolia var. curvifolia*	[31]
Apigenin **(10)**	C_15_H_10_O_5_	269	Capitula	*P. hilairei*	[32]
			Capitula	*L. curvifolia*	[40]
			Capitula	*L. flavescens*	[45]
			Capitula	*E. ligulatum*	[55]
			Capitula	*Syngonanthus suberosus* *S. dealbatus* *E. ligulatum* *L. spiralis*	[56]
			Leaves	*L. spiralis*	[56]
			Aerial part	*E. buergerianum*	[66]
Apigenin-4′-*O*-glucoside **(11)**	C_21_H_20_O_10_	431	Capitula	*P. hilairei*	[32]
Luteolin **(12)**	C_15_H_10_O_6_	285	Capitula	*P. planifolius*	[32]
			Capitula	*L. curvifolia*	[40]
			Aereal parts	*S. bisulcatus*	[49]
			Capitula	*L. flavescens*	[45]
			Capitula	*E. ligulatum*	[55]
			Scapes	*S. nintens*	[57]
			Scapes	*S. macrolepis*	[59]
			Aerial part	*S. nitens*	[33]
			Scapes	*Syngonanthus* spp.	[63]
3-methoxyquercetin **(13)**	C_16_H_12_O_7_	315	Capitula	*P. bifrons*	[32]
3-methoxyquercetagetin **(14)**	C_16_H_12_O_8_	331	Capitula	*P. planifolius*	[32]
			Capitula	*P. robustus*	[32]
3-methoxypatuletin **(15)**	C_17_H_14_O_8_	345	Capitula	*P. planifolius*	[32]
Apigenin-7-*O*-galactoside **(16)**	C_22_H_22_O_9_	429	Leaves	*S. suberosus*	[34]
Apigenin-7-*O*-glucoside **(17)**	C_22_H_22_O_9_	429	Leaves	*S. fuscescens* *S. eriopus*	[34]
			Capitula	*S. suberosus* *S. dealbatus* *E. ligulatum* *L. spiralis*	[56]
			Leaves	*L. spiralis*	[56]
			Capitula	*E. australe*	[62]
			Aerial part	*E. buergerianum*	[66]
Luteolin-7-*O*-arabinoside **(18)**	C_20_H_18_O_10_	417	Leaves	*S. laricifolius* *S. nitens*	[34]
Luteolin-7-*O*-galactoside **(19)**	C_21_H_20_O_11_	447	Leaves	*S. verticillatus* *S. elegans* *S. eriopus* *S. brasiliana* *S. xaranthemoides*	[34]
Luteolin-7-*O*-xyloside **(20)**	C_20_H_18_O_10_	417	Leaves	*S. aff. mucugensis* *S. xaranthemoides*	[34]
Luteolin-7-*O*-digalactoside **(21)**	-	-	Leaves	*S. elegans* *S. elegantulus* *S. eriopus* *S. xaranthemoides*	[34]
Luteolin-7-*O*-diglucoside **(22)**	-	-	Leaves	*S. anthemidiflorus* *S. arenarius* *S. niveus* *S. suberosus* *S. brasiliana* *S. xaranthemoides*	[34]
6-hydroxyluteolin-7-*O*-galactoside **(23)**	C_21_H_20_O_12_	463	Leaves	*S. verticillatus*	[34]
6-hydroxyluteolin-7-*O*-diarabinoside **(24)**	-	-	Leaves	*S. arenarius* *S. gracilis* *S. laricifolius*	[34]
6-hydroxyluteolin-7-*O*-digalactoside **(25)**	-	-	Leaves	*S. verticillatus* *S. macrolepis* *S. nitens*	[34]
6-hydroxyluteolin-7-*O*-diglucoside **(26)**	-	-	Leaves	*S. anthemidiflorus* *S. fuscescens* *S. helminthorryzus* *S. nitens*	[34]
6-methoxyquercetin-3-*O*-β-D-6′’-(*p*-coumaroyl)-glucopyranoside **(27)**	C_31_H_28_O_15_	639	Capitula	*P. polyanthus* *P. robustus* *P. denudatus*	[35]
			Capitula	*P. denudatus*	[52]
6-methoxykaempferol-3-*O*-β-D-6′’-(*p*-coumaroyl)-glucopyranoside **(28)**	C_31_H_28_O_14_	623	Capitula	*P. hilairei* *P. robustus* *P. ramosus* *P. denudatus*	[35]
			Capitula	*P. ramosus*	[39]
			Capitula	*P. macrocephalus*	[52]
			Flowers	*P. geniculatus*	[61]
			Capitula	*P. chiquitensis*	[64]
Quercetagetin-7-*O*-glucopyranoside **(29)**	C_21_H_20_O_13_	479	Capitula	*P. polyanthus* *P. hilairei* *P. ramosus*	[35]
			Capitula	*P. bromelioides*	[38]
			Capitula	*P. denudatus*	[52]
Patuletin-3-*O*-glucopyranoside **(30)**	C_22_H_22_O_13_	493	Capitula	*P. polyanthus* *P. robustus*	[35]
			Capitula	*P. macrocephalus* *P. denudatus*	[52]
			Aerial parts	*P. polyanthus*	[42]
			Capitula	*P. macropodus*	[43]
			whole plant	*E. buergerianum*	[53]
			Flowers	*P. geniculatus*	[61]
			Aerial parts	*Tonina fluviatilis*	[1]
			Capitula/seeds	*E. sexangulare* *E. buergerianum* *E. cinereum* *E. faberi*	[67]
			-	*P. brachypy*	[22]
Patuletin-3-*O*-rutinoside **(31)**	C_28_H_32_O_17_	639	Capitula	*P. polyanthus* *P. hilairei*	[35]
			Aerial parts	*P. polyanthus*	[42]
			Capitula/seeds	*E. buergerianum* *E. cinereum* *E. faberi*	[67]
6-methoxykaempferol **(32)**	C_16_H_12_O_7_	315	Capitula	*P. hilairei* *P. ramosus* *P. denudatus*	[35]
			Capitula	*P. hilairei* *P. vellozioides*	[38]
			Capitula	*P. chlorocephalus* *P. macrocephalus* *P. denudatus*	[52]
			-	*P. brachypy* *P. vellozioides* *P. latipes*	[22]
			-	*P. macropodus*	[22]
6-methoxykaempferol-3-*O*-glucopyranoside **(33)**	C_22_H_22_O_12_	477	Capitula	*P. hilairei* *P. robustus* *P. ramosus* *P. denudatus*	[35]
			Capitula	*P. hilairei* *P. bromelioides*	[38]
			Capitula	*P. ramosus*	[39]
			Capitula	*P. denudatus*	[52]
			Flowers	*P. geniculatus*	[61]
			Capitula	*P. acanthophyllus*	[65]
			-	*P. vellozioides* *P. latipes* *P. planifolius*	[22]
7-methoxyquercetagetin **(34)**	C_16_H_12_O_8_	331	Leaves	*P. latipes* *P. vellozioides*	[24]
			-	*P. bromelioides*	[22]
			Scapes	*P. latipes*	[36]
			Leaves	*P. vellozioides* *P. latipes*	[52]
7-methoxyquercetagetin-3-*O*-β-D-glucopyranoside **(35)**	C_22_H_22_O_13_	493	Leaves	*P. latipes* *P. vellozioides*	[24]
			Leaves	*P. vellozioides* *P. latipes*	[52]
			Aerial parts	*T. fluviatilis*	[1]
			-	*P. planifolius*	[22]
7-methoxyquercetagetin-4′-*O*-β-D-glucopyranoside **(36)**	C_22_H_22_O_13_	493	Leaves	*P. latipes*	[24]
			Leaves	*P. vellozioides* *P. latipes*	[52]
			Scapes	*P. latipes*	[36]
7-methoxyquercetagetin-3-*O*-cellobioside **(37)**	C_28_H_32_O_18_	655	Leaves	*P. latipes* *P. vellozioides*	[24]
7-methoxyquercetagetin-3-*O*-neohesperidoside **(38)**	C_28_H_32_O_17_	639	Leaves	*P. vellozioides*	[24]
			Aerial parts	*P. polyanthus*	[42]
7-methoxyquercetagetin-3-*O*-[2-*O*-caffeoyl-β-D-glucopyranosyl-(1-2)-*O*-β-D-glucuronopyranoside **(39)**	C_37_H_36_O_22_	831	Leaves	*P. latipes*	[24]
Patuletin-3-*O*-β-D-rutinoside **(40)**	C_28_H_32_O_17_	639	Capitula	*P. hilairei*	[38]
			Leaves	*P. vellozioides* *P. latipes*	[52]
			whole plant	*E. buergerianum*	[53]
5,7,4′-trihydroxy-6,3′-dimethoxyflavone **(41)**	C_17_H_14_O_7_	329	Capitula	*P. bromelioides*	[38]
			Capitula	*P. argenteus* var. *argenteus*	[52]
			Leaves	*P. argenteus* var. *argenteus*	[41]
			Capitula	*E. australe*	[62]
			Aerial part	*E. buergerianum*	[66]
			-	*L. curvifolia*	[22]
5,7,4′-trihydroxy-6,3′-dimethoxyflavonol **(42)**	C_17_H_14_O_8_	345	Capitula	*P. bromelioides*	[38]
5,3′-dihydroxy-7-4′,5′-trimethoxyisoflavone **(43)**	C_18_H_16_O_7_	343	Capitula	*L. curvifolia*	[40]
3′,4′,5,6-tetrahydroxy-7-methoxyflavone **(44)**	C_16_H_12_O_7_	315	Capitula	*L. flavescens*	[40]
			Leaves	*L. spiralis*	[60]
3′,4′,5-trihydroxy-7-methoxy-6-*C*-glucopyranosylflavone **(45)**	C_22_H_22_O_11_	461	Capitula	*L. flavescens*	[40]
			Scapes	*S. nintens*	[57]
			Leaves	*L. spiralis*	[60]
			Scapes	*S. macrolepis*	[59]
			Scapes	*Syngonanthus* spp.	[63]
			Aerial part	*E. buergerianum*	[66]
4′,5-dihydroxy-3′,6-dimethoxy-7-*O*-β-D-glucopyranosylflavone **(46)**	C_23_H_24_O_11_	475	Capitula	*L. flavescens*	[40]
			Capitula	*P. argenteus* var. *argenteus*	[52]
			Leaves	*P. argenteus* var. *argenteus*	[41]
			whole plant	*E. buergerianum*	[53]
			Capitula	*E. australe*	[62]
3′,4′,5,7-tetrahydroxy-6-*C*-glucopyranosylflavone **(47)**	C_21_H_20_O_11_	447	Capitula	*L. flavescens*	[40]
			Scapes	*S. nintens*	[57]
			Leaves	*L. spiralis*	[60]
			Scapes	*S. macrolepis*	[59]
			-	*S. bisulcatus*	[22]
3′,4′,5,6,7,8-hexahydroxyflavone **(48)**	C_15_H_10_O_8_	317	Capitula	*L. flavescens*	[40]
5,6,7,8,3′,4′-hexahydroxyflavonol **(49)**	C_15_H_10_O_9_	333	Aerial parts	*P. polyanthus*	[42]
			Capitula	*P. macropodus*	[43]
6-methoxyquercetin-3-*O*-(6′’-*E*-feruloyl)-β-D-glucopyranoside **(50)**	C_32_H_30_O_16_	669	Aerial parts	*P. polyanthus*	[42]
			whole plant	*E. buergerianum*	[53]
6-methoxyquercetagetin-3-*O*-β-D-glucopyranosyl-(1→4)-glucopyranoside **(51)**	C_28_H_32_O_18_	655	Capitula	*P. macropodus*	[43]
(2*S*)-3′,4′-methylenedioxy-5,7-dimethoxyflavan **(52)**	C_18_H_18_O_5_	313	Capitula	*E. buergerianum*	[44]
Hispidulin-7-(6-*E*-*p*-coumaroyl)-β-D-glucopyranoside **(53)**	C_31_H_28_O_13_	607	Capitula	*E. buergerianum*	[44]
			Capitula	*E. australe*	[62]
			Aerial part	*E. buergerianum*	[66]
Hispidulin **(54)**	C_16_H_12_O_6_	299	Capitula	*E. buergerianum*	[44]
			Capitula	*E. ligulatum*	[47]
			Capitula	*L. flavescens*	[45]
			whole plant	*E. buergerianum*	[53]
			Capitula	*E. ligulatum*	[55]
			Capitula	*E. australe*	[62]
			Aerial part	*E. buergerianum*	[66]
			Capitula/seeds	*E. sexangulare* *E. buergerianum* *E. cinereum* *E. faberi*	[67]
Hispidulin 7-*O*-glucoside **(55)**	C_22_H_22_O_11_	461	Capitula	*E. buergerianum*	[44]
			Capitula	*E. ligulatum*	[47]
			Capitula	*E. ligulatum*	[51]
			whole plant	*E. buergerianum*	[53]
			Capitula	*E. australe*	[62]
			Aerial part	*E. buergerianum*	[66]
			Capitula/seeds	*E. sexangulare* *E. buergerianum* *E. cinereum* *E. faberi*	[67]
3′,4′,5,6,7-pentahydroxyflavone **(56)**	C_15_H_10_O_7_	301	Capitula	*P. microphyllus*	[46]
			Scapes	*S. nintens*	[57]
			Scapes	*S. macrolepis*	[59]
			Scapes	*Syngonanthus* spp.	[63]
			-	*P. robustus* *P. planifolius* *P. flavescens*	[22]
3′,4′,5,6-tetrahydroxy-7-*O*-β-D-glucopyranosilflavone **(57)**	C_21_H_20_O_12_	463	Capitula	*P. microphyllus*	[46]
			Aereal parts	*S. bisulcatus*	[49]
			-	*P. planifolius*	[22]
			-	*P. flavescens*	[22]
6-methoxyluteolin-7-*O*-β-D-allopyranoside **(58)**	C_22_H_22_O_12_	477	Capitula	*E. ligulatum*	[47]
			Capitula	*E. ligulatum*	[48]
6-methoxyapigenin-7-*O*-β-D-allopyranoside **(59)**	C_22_H_22_O_11_	461	Capitula	*E. ligulatum*	[47]
			Capitula	*E. ligulatum*	[48]
			Capitula	*E. ligulatum*	[51]
			Capitula	*E. ligulatum*	[55]
6,4′-dimethoxyapigenin-7-*O*-β-D-glucopyranoside **(60)**	C_23_H_24_O_11_	475	Capitula	*E. ligulatum*	[47]
5,7,8,3′,4′-pentahydroxy-6-methoxyquercetin **(61)**	C_16_H_12_O_9_	347	Capitula	*E. ligulatum*	[47]
5-hydroxy-7,4′-dimethoxy-6-*C*-β-D-glucopyranosylflavone **(62)**	C_23_H_24_O_10_	459	Aereal parts	*S. bisulcatus*	[49]
isovitexin (5,7,4′-trihydroxy-6-*C*-β-D-glucopyranosylflavone) **(63)**	C_21_H_20_O_10_	431	Aereal parts	*S. bisulcatus*	[49]
lutonarin (5,3′,4′-trihydroxy-6-*C*-7-*O*-β-D-glucopyranosylflavone) **(64)**	C_27_H_30_O_16_	609	Aereal parts	*S. bisulcatus*	[49]
Xeractinol **(65)**	C_21_H_22_O_12_	465	Leaves	*P. argenteus* var. *argenteus*	[50]
6,4′-dimethoxyquercetin-3-O-β-D-6′’-[3,4,5-trihydroxy(*E*)-cinnamoyl]- glucopyranoside **(66)**	C_32_H_32_O_17_	687	Capitula	*E. ligulatum*	[51]
7-methoxyquercetagetin-3-*O*-glucopyranosyl-rhamnoside **(67)**	-	-	Leaves	*P. vellozioides* *P. latipes*	[52]
			Flowers	*P. geniculatus*	[61]
7-methoxyquercetagetin-3-*O*-[ caffeoyl]-glucopyranosyl-glucopyranoside **(68)**	-	-	Leaves	*P. vellozioides* *P. latipes*	[52]
6,3′-dimethoxyquercetin-7-*O*-β-D-glucopyranoside **(69)**	C_23_H_24_O_13_	507	Capitula	*P. macrocephalus*	[52]
			-	*P. bromelioides*	[22]
Apigenin-6-*C*-8-C-glucopyranoside **(70)**	C_27_H_30_O_15_	593	-	*P. planifolius*	[22]
7,3′-dihydroxy-5,4′,5′-trimethoxyisoflavone **(71)**	C_18_H_16_O_7_	343	whole plant	*E. buergerianum*	[53]
			Capitula/seeds	*E. buergerianum* *E. cinereum* *E. faberi*	[67]
Patuletin-3-*O*-[2-*O*-*E*-feruloyl-β-D-glucopyranosyl-(1→6)-β-D-glucopyranoside] **(72)**	C_38_H_40_O_21_	831	whole plant	*E. buergerianum*	[53]
Patuletin-3-*O*-[β-D-glucopyranosyl-(1→6)-2-*O*-*E*-caffeoyl-β-D-glucopyranosyl-(1→6)-β-D-glucopyranoside] **(73)**	C_43_H_48_O_26_	979	whole plant	*E. buergerianum*	[53]
			Capitula/seeds	*E. buergerianum* *E. cinereum* *E. faberi*	[67]
5,7,3′-trihydroxy-6,4′,5′-trimethoxyisoflavone **(74)**	C_18_H_16_O_7_	343	whole plant	*E. buergerianum*	[53]
Gerontoisoflavone A **(75)**	C_16_H_12_O_6_	299	whole plant	*E. buergerianum*	[53]
6-methoxyquercetin-7-*O*-β-D-glucopyranosyl-(1→6)-β-D-glucopyranoside **(76)**	C_28_H_32_O_18_	655	Capitula	*E. ligulatum*	[55]
			Capitula	*P. chiquitensis*	[64]
6-methoxyluteolin-7-*O*-β-D-glucopyranosyl-(1→6)-β-D-glucopyranoside **(77)**	C_28_H_32_O_17_	639	Capitula	*E. ligulatum*	[55]
6-methoxyquercetin-7-*O*-(6”‘-vanilloyl)-β-D-glucopyranosyl-(1→6)-β-D-glucopyranoside **(78)**	C_35_H_38_O_20_	777	Capitula	*E. ligulatum*	[55]
6,4′-dimethoxyquercetin-7-*O*-β-D-glucopyranosyl-(1→6)-β-D-glucopyranoside **(79)**	C_29_H_34_O_18_	669	Capitula	*E. ligulatum*	[55]
6-methoxyquercetin-7-*O*-β-D-glucopyranoside **(80)**	C_22_H_22_O_13_	493	Capitula	*E. ligulatum*	[55]
			Scapes	*P. chiquitensis*	[64]
6-methoxyquercetin-*O*-diglycosylrhamnoside **(81)**	-	-	Capitula	*E. ligulatum*	[56]
Rutin **(82)**	C_27_H_30_O_16_	609	Capitula	*S. suberosus* *S. dealbatus* *E. ligulatum* *L. spiralis*	[56]
			Leaves	*L. spiralis*	[56]
5,7,4′,5′-tetrahydroxy-3′-methoxy-6-*C*- β-D-glucopyranosylflavone **(83)**	C_22_H_22_O_12_	477	Scapes	*S. nintens*	[57]
5,4′,5′-trihydroxy-7,3′-dimethoxy-6-*C*- β-D-glucopyranosylflavone **(84)**	C_23_H_24_O_12_	491	Scapes	*S. nintens*	[57]
3′,4′,5-trihydroxy-7-methoxy-6-*C*-glucopyranosylflavanone **(85)**	C_22_H_24_O_11_	463	Scapes	*S. nintens*	[57]
5,4′,5′-trihydroxy-7,3′-dimethoxy-8-*C*- β-D-glucopyranosylflavone **(86)**	C_23_H_24_O_12_	491	Scapes	*S. nintens*	[57]
3′,4′,5-trihydroxy-7-methoxy-8-*C*-glucopyranosylflavone **(87)**	C_22_H_22_O_11_	461	Scapes	*S. nintens*	[57]
			Leaves	*L. spiralis*	[60]
			Scapes	*Syngonanthus* spp.	[63]
4′,5-dihydroxy-3′,7-dimethoxy-6-*C*-glucopyranosylflavone **(88)**	C_23_H_24_O_11_	475	Scapes	*S. nintens*	[57]
			Scapes	*S. macrolepis*	[59]
			Scapes	*Syngonanthus* spp.	[63]
3′,4′,5-trihydroxy-7-methoxy-8-*C*-glucopyranosylflavanone **(89)**	C_22_H_24_O_11_	463	Scapes	*S. nintens*	[57]
4′,5,7-trihydroxy-8-*C*-glucopyranosylflavanone **(90)**	C_21_H_22_O_10_	433	Scapes	*S. nintens*	[57]
4′-methoxyluteolin-6-*C*-β-D-glucopyranoside **(91)**	C_22_H_22_O_11_	461	Leaves	*L. spiralis*	[60]
6-methoxykaempferol-3-*O*-(2,3-di-*O*-acetyl)-α-L-rhamnopyranosyl-(1→6)-β-D-glucopyranoside **(92)**	C_32_H_36_O_18_	707	Flowers	*P. geniculatus*	[61]
6-methoxykaempferol-3-*O*-(2-*O*-acetyl)-α-L-rhamnopyranosyl-(1→6)-β-D-glucopyranoside **(93)**	C_30_H_34_O_17_	665	Flowers	*P. geniculatus*	[61]
6-methoxykaempferol-3-*O*-(4-*O*-acetyl)-α-L-rhamnopyranosyl-(1→6)-β-D-glucopyranoside **(94)**	C_30_H_34_O_17_	665	Flowers	*P. geniculatus*	[61]
6-methoxykaempferol-3-*O*-α-L-rhamnopyranosyl-(1→6)-β-D-glucopyranoside **(95)**	C_28_H_32_O_16_	623	Flowers	*P. geniculatus*	[61]
6-methoxyquercetin-3-*O*-(2-*O*-acetyl)-α-L-rhamnopyranosyl-(1→6)-β-D-glucopyranoside **(96)**	C_30_H_34_O_18_	681	Flowers	*P. geniculatus*	[61]
6-methoxyquercetin-3-*O*-(4-*O*-acetyl)-α-L-rhamnopyranosyl-(1→6)-β-D-glucopyranoside **(97)**	C_30_H_34_O_18_	681	Flowers	*P. geniculatus*	[61]
6-methoxyquercetin-3-*O*-(2,4-di-*O*-acetyl)-α-L-rhamnopyranosyl-(1→6)-β-D-glucopyranoside **(98)**	C_32_H_36_O_19_	723	Flowers	*P. geniculatus*	[61]
6-methoxykaempferol-3-*O*-α-L-rhamnopyranosyl-(1→2)-β-D-glucopyranoside **(99)**	C_28_H_32_O_16_	623	Flowers	*P. geniculatus*	[61]
6-methoxykaempferol-3-*O*-(6-*E*-feruloyl)-β-D-glucopyranoside **(100)**	C_32_H_30_O_15_	653	Flowers	*P. geniculatus*	[61]
Geniculatin **(101)**	C_62_H_56_O_28_	1247	Flowers	*P. geniculatus*	[61]
Eriocaulin A **(102)**	C_17_H_16_O_5_	299	Capitula	*E. australe*	[62]
3′,4′-methylenedioxyorobol **(103)**	C_16_H_10_O_6_	297	Capitula	*E. australe*	[62]
Iristectorigenin A **(104)**	C_17_H_14_O_7_	329	Capitula	*E. australe*	[62]
			Capitula/seeds	*E. sexangulare* *E. buergerianum* *E. cinereum* *E. faberi*	[67]
Irigenin **(105)**	C_18_H_16_O_8_	359	Capitula	*E. australe*	[62]
			Capitula/seeds	*E. buergerianum* *E. cinereum* *E. faberi*	[67]
Eriocauloside A **(106)**	C_31_H_28_O_12_	591	Capitula	*E. australe*	[62]
Eriocauloside B **(107)**	C_32_H_30_O_14_	637	Capitula	*E. australe*	[62]
Eriocauloside C **(108)**	C_30_H_28_O_14_	611	Capitula	*E. australe*	[62]
Hispidulin-7-*O*-β-D-(6-*O*-feruloyl)-glucopyranoside **(109)**	C_32_H_30_O_14_	637	Capitula	*E. australe*	[62]
Flavanonol-di-*O*-hexose **(110)**	-	627	Scapes	*P. chiquitensis*	[64]
Quercetin-3-*O*-di-hexose **(111)**	C_27_H_30_O_17_	625	Capitula	*P. chiquitensis*	[64]
			Capitula/seeds	*E. cinereum*	[67]
6-hydroxyquercetin-3-*O*-di-hexose **(112)**	C_27_H_30_O_18_	641	Scapes	*P. chiquitensis*	[64]
4′-methoxyapigenin-7-*O*-(3-galloyl)-α-D-arabinopyranosyl-(2→1)-apiofuranosyl-(3→1)-α-D-arabinopyranoside **(113)**	C_38_H_40_O_21_	831	Scapes	*P. chiquitensis*	[64]
6,3′-dimethoxyquercetin-7-*O*-β-D-glucopyranosyl-(6→1)-*O*-β-D-glucopyranoside **(114)**	C_29_H_34_O_18_	669	Capitula	*P. chiquitensis*	[64]
6-hydroxy-7,3,4-trimethoxyflavanonol **(115)**	C_18_H_18_O_8_	361	Scapes	*P. chiquitensis*	[64]
6-metoxykaempferol-3-*O*-hexose-*O*-pentose **(116)**	-	577	Scapes	*P. chiquitensis*	[64]
6-hydroxy-7-methoxyquercetin-3-*O*-pentose **(117)**	C_21_H_20_O_12_	463	Scapes	*P. chiquitensis*	[64]
7-methoxyquercetin-*O*-hexose **(118)**	C_22_H_24_O_15_	477	Capitula	*P. chiquitensis*	[64]
6-hydroxy-7-4-dimethoxyquercetin-3-*O*-hexose **(119)**	C_23_H_24_O_13_	507	Capitula	*P. chiquitensis*	[64]
6,3′-dimethoxyquercetin-3-*O*-β-D-6′’-(*p*-coumaroyl)-glucopyranoside **(120)**	C_32_H_30_O_15_	653	Capitula/scapes	*P. chiquitensis*	[64]
6,7-dimethoxyquercetin-3-*O*-β-D-glucopyranoside **(121)**	C_23_H_24_O_13_	507	Aerial parts	*T. fluviatilis*	[1]
6-methoxykaempferol-3-*O*-(6”-*p*-coumaroyl)-β-D-glucopyranosyl-7-*O*-β-D-glucopyranoside **(122)**	C_37_H_38_O_19_	785	Capitula	*P. acanthophyllus*	[65]
6-methoxykaempferol-3-7-di-*O*-β-D-glucopyranoside **(123)**	C_28_H_32_O_17_	639	Capitula	*P. acanthophyllus*	[65]
Kaempferol-7-*O*-β-D-6′’-(*p*-coumaroyl)-glucopyranoside **(124)**	C_30_H_26_O_12_	577	Aerial part	*E. buergerianum*	[66]
Patuletin-3-*O*-hexosyl-hexosyl-hexoside **(125)**	C_34_H_42_O_23_	817	Capitula/seeds	*E. buergerianum* *E. cinereum* *E. faberi*	[67]
Quercetagetin-3-*O*-hexoside **(126)**	C_21_H_20_O_13_	479	Capitula/seeds	*E. sexangulare*	[67]
Quercetin-3-*O*-hexosyl-pentoside **(127)**	C_26_H_30_O_17_	613	Capitula/seeds	*E. cinereum*	[67]
7-methoxy-hesperetin **(128)**	C_17_H_16_O_6_	315	Capitula/seeds	*-*	[67]
5,4′-dihydroxy-6,3′-dimethoxyflavone **(129)**	C_17_H_14_O_6_	313	Capitula/seeds	*E. sexangulare* *E. buergerianum* *E. cinereum*	[67]
Gerontoisoflavone A **(130)**	C_17_H_14_O_6_	313	Capitula/seeds	*E. cinereum* *E. faberi*	[67]

MF = molecular formula.

#### 2.1.2. Naphthopyranones

Naphthopyranone (1H-naphtho-[2,3-c]pyran-1-one) consists of a linear tricyclic system, with two cycles based in naphthalene ring fused to δ-lactone in the third ring. They are widely distributed in nature, having been isolated from fungi, bacteria, lichen, and plants. Many naphthopyranones tested showed biological activities such as antibiotic, antifungal, antimalarial, antioxidant, cytotoxic, immunoregulatory, and others [68].

After the isolation of a new compound named as paepalantine (**131**) from a chloroform extract of *Paepalanthus bromelioides* capitula [69], many other studies were performed with species from *Paepalanthus* genus.



Besides *Paepalanthus bromelioides*, compound **131**, paepalantine, was also isolated from *P. vellozioides* following Vilegas et al.’s (1990) separation procedure. Mutagenic and cytotoxic activities were tested against *Salmonella typhimurium* TA100, TA98, and TA102, and McCoy cells using the Neutral Red (NR) and microculture tetrazolium (MTT) techniques, respectively. Paepalantine showed mutagenic effect both in the absence and the presence of metabolic activation and also showed a IC_50_ equivalent to 30 and 38 mg mL^−1^ for NR and MTT, respectively [70]. In 1999, Tavares et al. [71] evaluated the clastogenic effect of compound **131**, with negative result, but observed significant cytotoxic activity when testing the compound in vitro and in vivo mammalian cell systems. The intestinal anti-inflammatory activity of compound **131** was also evaluated, resulting in paepalantine significantly attenuating the colonic damage induced by trinitrobenzenesulphonic acid (TNBS) both when colonic mucosa is intact or when the mucosa is recovering after an initial insult [72]. A study on the influence of dimethylsulfoxide (DMSO) in its antioxidant activity was also performed showing that DMSO significantly interfered with the hypochlorous acid (HOCl) assay, but propylene glycol may be the solvent of choice for paepalantine [73].

Paepalantine was found in capitula and scape of *Paepalanthus giganteus* [33] and also in *P. speciosus* and *P. microphyllus* [22]. Additionally, Vilegas et al. isolated glycoside paepalantine-9-*O*-β-D-glucopyranoside (**132**) and paepalantine-9-*O*-β-D-allopyranosyl(1→6)glucopyranoside (**133**) for the first time in 1998 in ethanolic extract from capitula of *P. bromelioides* [74]. The ethanolic extract, after partition between *n*-BuOH and water, was submitted to a DCCC fractionation, which produced **131**, **132**, and **133**. Compound **132** appeared as a yellow amorphous powder and compound **133** as yellow needles, but both appeared as yellowish-green spots on TLC.



Compounds **131** and **132** were also isolated from *Paepalanthus acanthophyllus*. The supernatant of methanol extract was fractioned in Sephadex LH-20 column and the fractions were filtered in SPE RP-18 cartridge. After chromatographic profile by HPLC-PDA, a chosen fraction was submitted to separation in a semipreparative HPLC-PDA, leading to isolation of both **131** and **132** [65]. Compound **132** was also isolated from *P. planifolius* [75].

When evaluating the cytotoxicity of the three naphthopyranones against McCoy cells using neutral red assays [76], **131–133** showed a significant cytotoxic index when compared with the IC_50_ value of cisplatin, a cytotoxic substance used in antineoplasic therapy. When tested for antimicrobial activity using a spectrophotometric microdilution technique, **131** was active against *Syngonanthus aureus*, *S. epidermidis*, and *Eriocaulon faecalis* whereas **132** and **133** proved ineffective against all microorganisms tested [77].

Leitão et al. (2002) also isolated compounds **131–133** in the same *Paepalanthus bromelioides* extract by HSCCC and evaluated their antioxidant activity. As a result, **131** showed good antioxidant activity in the DPPH radical assay [78]. Moreira et al. (2013) also isolated compound **133** and assessed the antimycobacterial activity using the Alamar Blue^TM^ (MABA) method disqualifying the naphthopyranone **133** as a promising candidate against *Mycobacterium tuberculosis* and *M. avium* [37].

Besides Paepalanthus bromelioides, compounds **132** and **133** were also isolated from P. hilairei, P. vellozioides, P. latipes [38], P. robustus, P. ramosus [35], P. macrocephalus, P. denudatus [52], P. speciosus, P. microphyllus [22], and the methanolic extract of Eriocaulon ligulatum capitula [47].

Compound 9,10-dihydroxy-7-methoxy-3-methyl-1H-naphtho-[2,3c]-pyran-1-one-9-*O*-β-D-allopyranosyl(1→6)glucopyranoside (**133**) was isolated from *Paepalanthus macropodus* capitula after chromatographing the EtOH extract on a Sephadex LH-20 column. Andrade et al. (2002) [43] identified the compound by spectroscopic methods and compared it to those previously reported.

The crude MeOH extracts of *Paepalanthus vellozioides* and *P. latipes* leaves, after purification with Amberlite XAD-2 followed by Sephadex LH-20 column, produced a crude glycosidic mixture, which was separated by reverse-phase HPLC, to yield pure compounds **134**, **135**, and **136**. Their structures were determined by spectroscopic and spectrometric techniques [23]. Additionally, the naphthopyranones **134** and **136** were identified in aerial parts of *Actinocephalus divaricatus* by high-resolution orbitrap mass spectrometry [79]. Compounds **134**, **135**, and **136** were also isolated in the whole *Eriocaulon buergerianum* species [53].



In 2001, the known 3,4-dihydro-10-hydroxy-7-methoxy-3-methyl-1H-3,4-dihydronaphtho [2,3c]pyran-1-one-9-*O*-β-D-allopyranosyl-(1→6)-β-D-glucopyranoside (**136**) was isolated from the ethanolic extract of *Paepalanthus planifolius* leaves. Additionally, the first naphthopyranone dimer (**137**), named planifolin, composed of monomeric portions of semi-vioxanthin and paepalantine linked by an ether bond, was isolated from the dichloromethane extract of *P. planifolius* capitula [80]. Cytotoxicity and mutagenic tests were performed with planifolin. The dimer showed a significant cytotoxic index (IC_50_ 12.83 µg mL^−1^) and mutagenic activity for TA100, TA98, and TA97a [81].



Three new naphthopyranone glycosides, named paepalantine-9-*O*-β-D-glucopyranosyl-(1→6)-β-D-glucopyranoside (**138**), paepalantine-9-*O*-α-L-arabinopyranosyl-(1→6)-β-D-glucopyranoside (**139**), and paepalantine-9-*O*-α-L-rhamnopyranosyl-(1→6)-β-D-glucopyranoside (**140**), and the known paepalantine-9-*O*-β-D-glucopyranoside (**132**) were isolated from aerial parts of *Paepalanthus microphyllus*. All compounds were tested in C8166 cells infected with HIV-1MN, showing disappointing activities [82].



The ethanol extract of aerial parts of *Paepalanthus bromelioides* was submitted to Sephadex LH-20 gel permeation column. The rechromatography of crude ethanolic extracts in silica gel resulted in isolation of two naphthopyranones, paepalantine-9-*O*-β-D-allopyranosyl-(1→6)-β-D-glycopyranoside (**133**), and paepalantine-9-*O*-β-D-glycopyranosyl-(1→6)-β-D-glycopyranoside (**138**) [37]. Both compounds had been isolated previously.

Another dimer was isolated by Coelho et al. (2000) [83]. This new naphthopyrone dimer (**141**) was isolated from *Paepalanthus bromelioides* capitula CH_2_Cl_2_ extract by chromatographic procedures. A colorimetric assay for cytotoxicity evaluation showed IC_50_ 55.9 µM. In 2004, Varanda et al. tested 8,8′-paepalantine dimer (**141**), **132**, and **133** mutagenicity in *Salmonella typhimurium* TA97a, TA98, TA100, and TA102 strains. Results showed mutagenic activity in TA97a strain treated with naphthopyranone **132** [84]. Amorim et al. (2018) also isolated **141** from *P. planifolius* [75].



Vioxanthin (**142**), a yellow–green powder, was isolated for the first time from acetone extracts of *Paepalanthus falcifolius* Koern*., P. albovaginatus* Alv. Silv., *P. argenteus* Koern., *P. cuspidatus* Alv. Silv., and *P. ramosus* Kunth. According to Provost and Garcia (1990), vioxanthin is very common in fungi of *Tricophyton* genus. However, those species studied were collected in different regions of Brazil and the plants showed no detectable contamination by microorganisms. Thus, this was the first isolation from a plant source [85]. Vioanthin was also found in *P. bromelioides* capitula CH_2_Cl_2_ extract and in *P. planifolius* capitula EtOAc extract [75]. Furthermore, effects of compounds **131**, **141**, and **142** on mitochondria were evaluated [86,87].



Eriocauline (**143**) is a naphthopyranone dimer isolated from *Eriocaulon ligulatum* capitula. The dichloromethane extracts were submitted to column chromatography on silica gel and their structure was identified by spectroscopic and spectrometric experiments [51].



Three other compounds, **144**, **145**, and **146**, were characterized from *Eriocaulon ligulatum* capitula. The methanolic extract was analyzed by liquid chromatography electrospray ionization multistage ion trap mass spectrometry (LC-ESI-IT-MS^n^). All three compounds were identified based on their fragmentation patterns in MS and in NMR spectra. This was the first report of **144** and **145** in *Eriocaulon* genus [55]. Additionally, compound **146** was isolated in the whole *E. buergerianum* [53] and in *Paepalanthus denudatus* and *P. speciosus* [22,52].



The ethanolic extract of *Paepalanthus macrocephalus* was analyzed by HPLC-ES-MS in an RP-18 column with MeCN/H_2_0 (30:70% + 1% HCOOH) as eluent system. The *m/z* 273 and the *m/z* 597 suggested the presence of compounds 5-desmethoxypaepalantine (**147**) and 5-desmethoxypaepalantine-9-*O*-β-D-glucopyranosyl(1→6)glucopyranosideo (**148**), respectively [22,52]. Compounds **147** and **148** were also isolated in the hydroethanolic extract of *Eriocaulon buergerianum* Koern. The EtOAc fraction yielded compound **147**, and the BuOH and H_2_O fraction yielded **148** [66].



*Paepalanthus geniculatus* Kunth. flowers were submitted to HPLC-ESI-MS^n^ analysis after the methanol extract showed radical-scavenging activity in the Trolox equivalent antioxidant capacity (TEAC) assay. Two known naphthopyranones (**149**, **150**) and two new ones (**151**, **152**) were isolated. In antioxidant assay, compounds **151** and **152** showed the highest TEAC values [61]. The naphthopyranones **151** and **152** were identified in aerial parts of *Actinocephalus divaricatus* by high-resolution orbitrap mass spectrometry [79].



The dichloromethane extract of *Paepalanthus diffissus* rhizomes led to a precipitate of a green solid residue, which was purified by chromatography and crystallization, providing a pure yellow needle of (+)-Semi-vioxanthin (**153**). Another fraction was chromatographed on a Sephadex LH-20 column giving a crystalline solid like pale-yellow needles, identified as vioxanthin (**142**) [88]. Additionally, Semi-vioxanthin (**153**) was isolated from *P. planifolius* capitula EtOAc extract [75].



The methanol extracts of *Paepalanthus chiquitensis* capitula and scapes were submitted to HPLC-ESI-IT-MS^n^ to characterization. In capitula, the naphthopyranones **132**, **146**, **154**, and **155** were identified, whereas, in scapes, **146** and **154** were identified [64]. Compound **155** was also isolated from *Actinocephalus divaricatus* (Körn.) Sano methanolic extract [79].



Studying *Paepalanthus planifolius* capitula EtOAc extract allowed the identification of paepalantine-9-*O*-β-D-glucopyranosyl (**132**), 1H-naphtho[2,3-c]pyran-1-one,9-[(6-*O*-β-D-glucopyranosyl-β-D-glucopyranosyl)oxy]-3,4-dihydro-10-hydroxy-7-methoxy-3-methyl (**135**), paepalantine dimer (**141**), vioxanthin (**142**), semi-vioxanthin-9-*O*-β-D-glucopyranoside (**145**), and semivioxanthin (**153**), as previously mentioned [75]. Furthermore, Amorim et al. (2018) [75] also identified 1H-naphtho[2,3-c]pyran-1-one,3,4-dihydro-9,10-dihydroxy-5,7-dimethoxy-3-methyl (**156**). A new naphthopyranone dimer named planifoliusin A (**157**) was also isolated. Other naphthopyranone dimers (**158–163**) were proposed by MS fragmentation patterns.



Qiao et al. (2012) [67], studying the chemical constituents of a Chinese herbal medicine Gu-Jing-Cao (*Eriocaulon buergerianum*) to identify adulterating species by HPLC-DAD-ESI-MS^n^, identified the naphthopyranones **146**, **148**, and **164** in *Eriocaulon sexangulare*; **145**, **148**, **153**, and **164**–**166** in *E. buergerianum*; **145**, **148**, **153**, and **164**–**166** in *E. cinereum*; and **148**, **153**, and **164**–**166** in *E. faberi*.



In short, they found thirty-six naphthopyranones isolated from one species of *Actinocephalus*, five species of *Eriocaulon,* and twenty-one species of *Paepalanthus*, being four aglycones, twenty-one glycones, and eleven dimers. The dimers were isolated in *Paepalanthus* and *Eriocaulon* at the positions C-8, C-9, and C-10. The glycosylation pattern occurs only at the C-9 position, with a binding between the naphthopyranone and α-glucose or β-glucose being very common. When the binding is between a naphthopyranone and a disaccharide, the glucoside unit starts with a glucose. The sugars forming disaccharides with glucose are glucose, rhamnose, arabinose, and allose.

Table 2 shows the names of naphthopyranones, the part of the plant where they were isolated, species, and authors.

**Table 2 molecules-27-07186-t002:** Naphthopyranones isolated from Eriocaulaceae species.

Compounds	MF	[M−H]^−^	Organ	Species	Reference
Paepalantine **(131)**	C_16_H_14_O_6_	301	Capitula	*Paepalanthus bromelioides*	[69]
			Capitula	*P. bromelioides*	[72]
			Capitula	*P. bromelioides*	[73]
			Capitula	*P. bromelioides*	[78]
			Capitula	*P. bromelioides*	[86]
			Capitula	*P. bromelioides*	[87]
			Capitula	*P. vellozioides*	[70]
			Capitula	*P. vellozioides*	[71]
			Capitula/ Scape	*P. giganteus*	[33]
			-	*P. speciosus*	[22]
			-	*P. microphyllus*	[22]
			Capitula	*P. acanthophyllus*	[65]
Paepalantine-9-*O*-β-D-glucopyranoside **(132)**	C_22_H_24_O_11_	463	Capitula	*P. bromelioides*	[74]
			Capitula	*P. bromelioides*	[78]
			Capitula	*P. bromelioides*	[38]
			Capitula	*P. acanthophyllus*	[65]
			Capitula	*P. planifolius*	[75]
			Capitula	*P. hilairei*	[38]
			Capitula	*P. hilairei*	[35]
			Capitula	*P. vellozioides*	[38]
			Capitula	*P. latipes*	[38]
			Capitula	*P. robustus*	[35]
			Capitula	*P. ramosus*	[35]
			Capitula	*P. denudatus*	[35]
			Capitula	*P. denudatus*	[52]
			Capitula	*P. macrocephalus*	[52]
			Capitula	*P. chiquitensis*	[64]
			Capitula	*Eriocaulon ligulatum*	[47]
			-	*P. speciosus*	[22]
			-	*P. microphyllus*	[22]
			Aerial part	*P. microphyllus*	[82]
Paepalantine-9-*O*-β-D-allopyranosyl-(1→6)-glucopyranoside **(133)**	C_28_H_34_O_16_	625	Capitula	*P. bromelioides*	[74]
			Capitula	*P. bromelioides*	[78]
			Capitula	*P. bromelioides*	[38]
			Aerial part	*P. bromelioides*	[37]
			Capitula	*P. hilairei*	[38]
			Capitula	*P. hilairei*	[35]
			Capitula	*P. vellozioides*	[38]
			Capitula	*P. latipes*	[38]
			Capitula	*P. robustus*	[35]
			Capitula	*P. ramosus*	[35]
			Capitula	*P. denudatus*	[35]
			Capitula	*P. denudatus*	[52]
			Capitula	*P. macrocephalus*	[52]
			Capitula	*P. macropodus*	[43]
			Capitula	*E. ligulatum*	[47]
			-	*P. speciosus*	[22]
			-	*P. microphyllus*	[22]
3,4-dihydro-10-hydroxy-7-methoxy-3-(*R*)-methyl-1H-3,4-dihydronaphtho-[2,3c]-pyran-1-one-9-*O*-β-D-glucopyranoside **(134)**	C_21_H_24_O_10_	435	Leaves	*P. vellozioides*	[23]
			Leaves	*P. latipes*	[23]
			Aerial parts	*Actinocephalus divaricatus*	[79]
			Whole plant	*E. buergerianum*	[53]
3,4-dihydro-10-hydroxy-7-methoxy-3-(*R*)-methyl-1H-3,4-dihydronaphtho-[2,3c]-pyran-1-one-9-*O*-β-D-glucopyranosyl-(1→6)-glucopyranoside **(135)**	C_27_H_34_O_15_	597	Leaves	*P. vellozioides*	[23]
			Leaves	*P. latipes*	[23]
			Whole plant	*E. buergerianum*	[53]
			Capitula	*P. planifolius*	[75]
3,4-dihydro-10-dihydroxy-7-methoxy-3-(*R*)-methyl-1H-3,4-dihydronaphtho-[2,3c]-pyran-1-one-9-*O*-β-D-allopyranosyl-(1→6)-glucopyranoside **(136)**	C_27_H_34_O_15_	597	Leaves	*P. vellozioides*	[23]
			Leaves	*P. latipes*	[23]
			Aerial parts	*A. divaricatus*	[79]
			Whole plant	*E. buergerianum*	[53]
			Leaves	*P. planifolius*	[80]
Planifolin **(137)**	C_31_H_26_O_10_	557	Capitula	*P. planifolius*	[80]
Paepalantine-9-O-β-D-glucopyranosyl-(1→6)-β-D-glucopyranoside **(138)**	C_28_H_34_O_16_	625	Aerial part	*P. microphyllus*	[82]
			Aerial part	*P. bromelioides*	[37]
Paepalantine-9-*O*-a-L-arabinopyranosyl-(1→6)-β-D-glucopyranoside **(139)**	C_27_H_32_O_15_	595	Aerial part	*P. microphyllus*	[82]
Paepalantine-9-O-a-L-rhamnopyranosyl-(1→6)-β-D-glucopyranoside **(140)**	C_28_H_34_O_15_	609	Aerial part	*P. microphyllus*	[82]
8,8′-paepalantine dimer **(141)**	C_32_H_28_O_12_	603	Capitula	*P. bromelioides*	[83]
			Capitula	*P. bromelioides*	[86]
			Capitula	*P. bromelioides*	[87]
			Capitula	*P. planifolius*	[75]
Vioxanthin **(142)**	C_30_H_26_O_10_	545	Collar	*P. falcifolius*	[85]
			Collar	*P. albovaginatus*	[85]
			Collar	*P. argenteus*	[85]
			Collar	*P. cuspidatus*	[85]
			Collar	*P. ramosus*	[85]
			Capitula	*P. bromelioides*	[86]
			Capitula	*P. bromelioides*	[87]
			Capitula	*P. planifolius*	[75]
			Rhizomes	*P.diffissus*	[88]
Eriocauline **(143)**	C_30_H_22_O_8_	509	Capitula	*E. ligulatum*	[51]
9,10-dihydroxy-7-methoxy-3-(*R*)-methyl-1H-naphtho[2,3c]pyran-1-one-9-*O*-β-D-allopyranosyl-(1→6)-glucopyranoside **(144)**	C_27_H_32_O_15_	595	Capitula	*E. ligulatum*	[38]
9,10-dihydroxy7-methoxy-3-(*R*)-methyl-1H-3,4-dihydronaphtho[2,3c]pyran-1-one-9-*O*-β-D-glucopyranoside **(145)**	C_21_H_24_O_10_	435	Capitula	*E. ligulatum*	[38]
			Capitula	*P. planifolius*	[75]
			Capitula/seeds	*E. buergerianum*	[67]
			Capitula/seeds	*E. cinereum*	[67]
9,10-dihydroxy-7-methoxy-3-(*R*)-methyl-1Hnaphtho[2,3c]pyran-1-one-9-*O*-β-D-glucopyranoside **(146)**	C_21_H_22_O_10_	433	Capitula	*E. ligulatum*	[38]
			Whole plant	*E. buergerianum*	[53]
			Capitula	*P. denudatus*	[52]
			Capitula	*P. chiquitensis*	[64]
			Scapes	*P. chiquitensis*	[64]
			Capitula/seeds	*E. sexangulare*	[67]
			-	*P. speciosus*	[22]
5-desmethoxypaepalantine **(147)**	C_15_H_12_O_5_	271	Capitula	*P. macrocephalus*	[52]
			Aerial part	*E. buergerianum*	[66]
5-desmethoxypaepalantine-9-*O*-β-D-glucopyranosyl-(1→6)-glucopyranosideo **(148)**	C_27_H_32_O_15_	595	Capitula	*P. macrocephalus*	[52]
			Aerial part	*E. buergerianum*	[66]
			Capitula/seeds	*E. sexangulare*	[67]
			Capitula/seeds	*E. cinereum*	[67]
			Capitula/seeds	*E. faberi*	[67]
Paepalantine-9-*O*-α-D-glucopyranosyl-(1→6)-α-D-glucopyranoside **(149)**	C_28_H_34_O_16_	625	Flowers	*P. geniculatus*	[61]
Paepalantine-9-*O*-β-D-allopyranosyl-(1→6)-α-D-glucopyranoside **(150)**	C_28_H_34_O_16_	625	Flowers	*P. geniculatus*	[61]
7,9,10-trihydroxy-5-methoxy-3-methyl-1H-naphtho[2,3-c]pyran-1-one-9-*O*-β-D-glucopyranosyl-(1→6)-β-D-glucopyranoside **(151)**	C_27_H_32_O_16_	611	Flowers	*P. geniculatus*	[61]
			Aerial parts	*A. divaricatus*	[79]
7,9,10-trihydroxy-5-methoxy-3-methyl-1H-naphtho[2,3-c]pyran-1-one-9-*O*-β-D-glucopyranoside **(152)**	C_21_H_22_O_11_	449	Flowers	*P. geniculatus*	[61]
			Aerial parts	*A. divaricatus*	[79]
(+)-Semi-vioxanthin **(153)**	C_15_H_14_O_5_	273	Rhizomes	*P.diffissus*	[88]
			Capitula	*P. planifolius*	[75]
			Capitula/seeds	*E. buergerianum*	[67]
			Capitula/seeds	*E. cinereum*	[67]
			Capitula/seeds	*E. faberi*	[67]
5-10-Ddihydroxy-7-methoxy-3-methyl-1Hnaphtho[2,3c]pyran-1-one-9-*O*-α-L-rhamnopyranosyl-(1→6)-*O*-β-D-glucopyranoside **(154)**	C_27_H_32_O_15_	595	Capitula	*P. chiquitensis*	[64]
			Scapes	*P. chiquitensis*	[64]
10-Hydroxy-5,7-dimethoxy-3-methyl-1H-naphtho[2,3c]pyran1-one-9-*O*-β-D-allopyranosyl-(1→6)-*O*-β-D-glucopyranoside **(155)**	C_28_H_34_O_16_	625	Capitula	*P. chiquitensis*	[64]
			Aerial parts	*A. divaricatus*	[79]
1H-naphtho[2,3-c]pyran-1-one,3,4-dihydro-9,10-dihydroxy-5,7-dimethoxy-3-methyl **(156)**			Capitula	*P. planifolius*	[75]
Planifoliusin A **(157)**	C_31_H_26_O_11_	573	Capitula	*P. planifolius*	[75]
Planifoliusin B **(158)**	C_31_H_28_O_11_	575	Capitula	*P. planifolius*	[75]
Planifoliusin C **(159)**	C_32_H_30_O_12_	605	Capitula	*P. planifolius*	[75]
Planifoliusin D **(160)**	C_32_H_24_O_10_	543	Capitula	*P. planifolius*	[75]
Planifoliusin E **(161)**	C_32_H_28_O_12_	603	Capitula	*P. planifolius*	[75]
Planifoliusin F **(162)**	C_30_H_22_O_10_	541	Capitula	*P. planifolius*	[75]
Planifoliusin G **(163)**	C_31_H_24_O_11_	571	Capitula	*P. planifolius*	[75]
Semivioxanthin-9-*O*-hexosyl-hexoside **(164)**	C_27_H_34_O_15_	597	Capitula/seeds	*E. sexangulare*	[67]
			Capitula/seeds	*E. buergerianum*	[67]
			Capitula/seeds	*E. cinereum*	[67]
			Capitula/seeds	*E. faberi*	[67]
(*R*)-Semivioxanthin-9-*O*-β-D-allosyl-(1→6)-glucoside **(165)**	C_27_H_34_O_15_	597	Capitula/seeds	*E. buergerianum*	[67]
			Capitula/seeds	*E. cinereum*	[67]
			Capitula/seeds	*E. faberi*	[67]
(*R*)-Semivioxanthin-9-*O*-β-D-glucosyl-(1→6)-glucoside **(166)**	C_27_H_34_O_15_	597	Capitula/seeds	*E. buergerianum*	[67]
			Capitula/seeds	*E. cinereum*	[67]
			Capitula/seeds	*E. faberi*	[67]

MF = molecular formula.

#### 2.1.3. Compounds Isolated from Eriocaulaceae Fungi

Living plants are colonized by plenty of micro-organisms [89]. Endophytic fungi are microorganisms that inhabit internal plant tissues and provide benefits to the host [90]. They may produce equal or analogous metabolites as their host plants with potential application in many agricultural and industrial segments, which increased many scientists’ interests in studying potential biologically active compounds [91].

Amorim et al. (2016) isolated endophytic fungi from *Paepalanthus planifolius* capitula, scapes, and leaves [92]. After treatment for fungus proliferation, fifteen single fungal strains were obtained and one of them was identified as *Anthostomella brabeji*. *A. brabeji* was submitted to liquid–liquid partition with ethyl acetate, resulting in a crude extract. The ethyl acetate extract was purified by semipreparative HPLC-DAD, leading to the isolation of four compounds: new natural product **167** (6.1 mg), **168** (19.4 mg), **169** (9.7 mg), and **170** (13.6 mg). The isolated compounds and EtOAC extract were assayed against the microorganisms *Candida albicans*, *Escherichia coli*, *Salmonella setubal*, and *Staphylococcus aureus*. Measured MIC values ranged from 31.25 μg mL^−1^ to 1000.0 μg mL^−1^.



Hilário et al. (2017), studying aerial parts of *Paepalanthus chiquitensis*, isolated the endophytic fungi *Fusarium fujikuroi* and found the ethyl acetate extract was bioactive against Gram-positive bacteria *Staphylococcus aureus*, Gram-negative *Escherichia coli* and *Salmonella setubal*, and the fluconazole-resistant yeast *Candida albicans*. This chemical study yielded an alkaloid 2-(4-butylpicolinamide) acetic acid (**171**) and three known compounds: fusaric acid (**172**), indole acetic acid (**173**), and terpestacin (**174**). The minimal inhibitory concentration of the extract, fusaric acid, and indole acetic acid had values from 125 to 1000 μg mL^−1^ [93].



Three years after the first study on metabolites isolated from endophytic fungi of *Paepalanthus planifolius*, Amorim et al. (2019) isolated three new benzaldehyde derivatives, sporulosaldeins A–C (**175–177**), and three new benzopyran derivatives, sporulosaldeins D–F (**178–180**) from *Paraphaeosphaeria* sp. F03 in *P. planifolius* leaves. All isolated compounds had their chemical structure elucidated by nuclear magnetic resonance experiments and high-resolution mass spectrometry analysis. The cytotoxic MCF-7 and LM3 cells and antimicrobial activities were also tested [94].



Continuing their studying on aerial parts, [95] isolated the endophytic fungus *Curvularia lunata* of *Paepalanthus chiquitensis* capitula. The EtOAc extract (1.3 g) obtained from *C. lunata* was fractionated by Sephadex LH-20 column chromatography and eluted with 100% MeOH. Those proceedings yielded two new spirocyclic γ-lactams (**183–184**), and the known triticones E (**181**) and F (**182**), 5-*O*-methylcurvulinic acid (**185**), curvulinic acid (**186**), and curvulin (**187**). The antimicrobial assay was performed, showing a good activity for triticones **181** and **182** against *Escherichia coli* (MIC 62.5 μg mL^−1^) [95].



In short, those compounds isolated from Eriocaulaceae family derived from acetylenic phenols, acids, alkaloids, benzaldehydes, benzopyran, chromanones, and sesterpenes. Most of them were isolated from endophytic fungi present in aerial parts of three different species of *Paepalanthus*. Table 3 shows that information.

**Table 3 molecules-27-07186-t003:** Compounds of endophytic fungi isolated from *Paepalanthus* species.

Compounds	MF	[M−H]^−^	Organ	Species	Reference
(+)-(6*R**,7*S**,8*R**)-6,7,8-trihydroxy- 2,2-dimethyl-5,6,7,8-tetrahydro-croman-4-one **(167)**	C_11_H_16_O_5_	227	Capítulo/ *Anthostomella brabeji*	*Paepalanthus planifolius*	[92]
6-hydroxy-2,2-dimethyl-5,6,7,8- tetrahydro-7,8-epoxychroman-4-one **(168)**	C_11_H_14_O_4_	209	Capítulo/ *Anthostomella brabeji*	*P. planifolius*	[92]
Sicaine **(169)**	C_11_H_10_O_2_	173	Capítulo/ *Anthostomella brabeji*	*P. planifolius*	[92]
Eutypinol **(170)**	C_12_H_12_O_2_	187	Capítulo/ *Anthostomella brabeji*	*P. planifolius*	[92]
2-(4-butylpicolinamide) acetic acid **(171)**	C_12_H_16_N_2_O_3_	235	Parte aérea/ *Fusarium fujikuroi*	*P. chiquitensis*	[93]
Fusaric acid **(172)**	C_10_H_13_NO_2_	178	Partes aéreas/ *Fusarium fujikuroi*	*P. chiquitensis*	[93]
Indoleacetic acid **(173)**	C_10_H_9_NO_2_	174	Partes aéreas/ *Fusarium fujikuroi*	*P. chiquitensis*	[93]
Terpestacine **(174)**	C_25_H_38_O_4_	401	Parte aérea/ *Fusarium fujikuroi*	*P. chiquitensis*	[93]
Sporulosaldeine A **(175)**	C_12_H_14_O_5_	237	Folhas/ *Paraphaeosphaeria sp.* F03	*P. planifolius*	[94]
Sporulosaldeine B **(176)**	C_12_H_12_O_5_	235	Folhas/ *Paraphaeosphaeria sp.* F03	*P. planifolius*	[94]
Sporulosaldeine C **(177)**	C_12_H_14_O_4_	221	Folhas/ *Paraphaeosphaeria sp.* F03	*P. planifolius*	[94]
Sporulosaldeine D **(178)**	C_12_H_12_O_4_	219	Folhas/ *Paraphaeosphaeria sp.* F03	*P. planifolius*	[94]
Sporulosaldeine E **(179)**	C_12_H_14_O_4_	221	Folhas/ *Paraphaeosphaeria sp.* F03	*P. planifolius*	[94]
Sporulosaldeine F **(180)**	C_13_H_14_O_5_	249	Folhas/ *Paraphaeosphaeria sp.* F03	*P. planifolius*	[94]
Triticones E **(181)**	C_14_H_19_NO_6_	296	Capitula/ *Curvularia lunata*	*P. chiquitensis*	[95]
Triticones F **(182)**	C_14_H_19_NO_6_	296	Capitula/ *Curvularia lunata*	*P. chiquitensis*	[95]
(1*S**,2*Z*,3*Z*,3′*S*,6*R**)-6,6′-dihydroxy-1′-methoxy-2-ropylidene-3′,4′-dihydro-2′H-spiro[cyclohexane-1,3′-cyclopenta[β]pyrrol]-3-ene-2′,5,5′(1′H)-trione **(183)**	C_16_H_17_NO_6_	318	Capitula/ *Curvularia lunata*	*P. chiquitensis*	[95]
(1*S**,2*Z*,3*Z*,3′*S*,6*S**)-6,6′-dihydroxy-1′-methoxy-2-ropylidene-3′,4′-dihydro-2′H-spiro[cyclohexane-1,3′-cyclopenta[β]pyrrol]-3-ene-2′,5,5′(1′H)-trione **(184)**	C_16_H_17_NO_6_	318	Capitula/ *Curvularia lunata*	*P. chiquitensis*	[95]
5-*O*-methylcurvulinic acid **(185)**	C_11_H_12_O_5_	223	Capitula/ *Curvularia lunata*	*P. chiquitensis*	[95]
Curvulinic acid **(186)**	C_10_H_10_O_5_	209	Capitula/ *Curvularia lunata*	*P. chiquitensis*	[95]
Curvulin **(187)**	C_12_H_14_O_5_	237	Capitula/ *Curvularia lunata*	*P. chiquitensis*	[95]

MF = molecular formula.

#### 2.1.4. Xanthones

A xanthone skeleton is planar, with a conjugated ring system A and B linked to a carbonyl group and an oxygen atom. They are classified as oxygenated xanthones, prenylated xanthones, xanthone glycosides, bis-xanthones, xanthonolignoids, miscellaneous xanthones, and polyphenolic xanthones are also divided into subclasses depending upon the degree of oxygenation. According to the literature, about 650 xanthones are known from natural sources and diverse biological activities are described for this class [96].

In Eriocaulaceae, seventeen xanthones were isolated, all hydroxylated and with at least one methoxyl group in their structure. Santos et al. (2001) described the first xanthone in the family [40]. Xanthones **188** (10.0 mg) and **189** (12.0 mg) were isolated from EtOH extract (2.0 g) of *Leiothrix curvifolia* and **190** (4.2 mg) of *Leiothrix flavescens*. The structure was characterized by spectroscopic methods, mainly NMR experiments, and by electrospray mass spectrometry. In 2003, Santos et al. investigated the antioxidant effect using TEAC and linoleic acid assays of the xanthones **188**, **189**, and **190**. The evaluation showed moderate antioxidant activity when compared with quercetin and BHT (2,6-di-tertbutyl-4-methoxyphenol) [97]. In 2011, Araújo et al. isolated xanthones **189** and **190** from methanolic extracts of *L. spiralis* Ruhland leaves. Antimicrobial activity was also tested against Gram-positive (*Staphylococcus aureus*, *Bacillus subtilis*, and *Enterococcus faecalis*), Gram-negative bacteria (*Escherichia coli*, *Pseudomonas aeruginosa*, *Salmonella setubal*, and *Helicobacter pylori*), and fungi (the yeasts *Candida albicans*, *C. tropicalis*, *C. krusei*, and *C. parapsilosis*). The best result for **189** was against all tested *Candida* strains [60].



Fang et al. (2008) isolated **191** (36.0 mg), **192** (52.0 mg), and **193** (60.0 mg) from *Eriocaulon buergerianum* EtOAc extract after successive separation with polyamide and silica gel chromatography. Their structures were determined by spectroscopic methods. All compounds were tested against the pathogenic bacteria *Staphylococcus aureus* (ATCC 25923), but those xanthones showed no significant antibacterial activity [53].



Pacífico et al. (2011), investigating flower Sephadex fractions from *Syngonanthus nitens*, isolated **193** (5.3 mg), **194** (11.2 mg), **195** (6.8 mg), **196** (5.1 mg), and **197** (10.2 mg), with compound **194** being described for the first time. All compounds had their chemical structures determined by NMR and ESI-MS^n^ experiments [57].

Oliveira et al. (2013), investigating methanol extract of *Syngonanthus* spp., isolated the mixture of **194/195**, **197**, **198**, and **199**. The authors evaluated their estrogenicity, mutagenic, and antimutagenic properties. The mixture of xanthones **194/195** and xanthone **198** had the best results in comparison with the *Syngonanthus nitens* extract. The most estrogenic concentration found for the **194/195** xanthone mixture was 1.5 µg well^−1^; for **197** it was 0.9 µg well^−1^; for **198** it was 1.00 µg well^−1^ and for **199** it was 20 µg well^−1^. None of the isolated compounds showed mutagenicity against the *Salmonella typhimurium* TA100, TA98, TA97a, and TA102 strains but all of them showed antimutagenic potential [63]. Xanthone **199**, 1,5,7-trihydroxy-3,6-dimethoxyxanthone, was identified in methanol extract of *S. nitens* capitula by performing co-injection experiments in HPLC-PDA [33].



Qiao et al. (2012), studying the chemical constituents of a Chinese herbal medicine Gu-Jing-Cao (*Eriocaulon buergerianum*) to identify adulterating species by HPLC-DAD-ESI-MS^n^, identified xanthone **191** and **192** in *E. faberi*, **195** in *E. buergerianum*, and **200** in *E. cinereum* and *E. faberi*. The xanthone **193** was also found, but the authors did not specify in which species [67].



Liang et al. (2018) isolated **192** and **201** from *Eriocaulon buergerianum* vines and both xanthones showed significant fatty acid synthase inhibition activity with half inhibitory concentration values of 12.5 and 13.6 μM, respectively [54].



Table 4 shows the names of xanthones, the part of the plant where they were isolated, species, and authors.

**Table 4 molecules-27-07186-t004:** Xanthones isolated in Eriocaulaceae species.

Compounds	MF	[M−H]^−^	Organ	Species	Reference
8-carboxymethyl-1,6-dihydroxy-3,5-dimethoxyxanthone **(188)**	C_17_H_14_O_8_	345	Capitula	*Leiothrix curvifolia*	[40]
8-carboxymethyl-1,5,6-trihydroxy-3-methoxyxanthone **(189)**	C_16_H_12_O_8_	331	Capitula	*L. curvifolia*	[40]
8-carboxymethyl-1,3,5,6-tetrahydroxyxanthone **(190)**	C_15_H_10_O_8_	317	Capitula	*L. flavescens*	[40]
1,3,6-trihydroxy-2,5,7-trimethoxyxanthone **(191)**	C_16_H_14_O_8_	333	whole plant	*Eriocaulon buergerianum*	[53]
			Capitula, seeds	*E. faberi*	[67]
1,3,6,8-tetrahydroxy-2,7-dimethoxyxanthone **(192)**	C_15_H_12_O_8_	319	whole plant	*E. buergerianum*	[53]
			vines	*E. buergerianum*	[54]
1,3,6,8-tetrahydroxy-2-methoxyxanthone **(193)**	C_14_H_10_O_7_	289	whole plant	*E. buergerianum*	[53]
			flowers	*Syngonanthus nitens*	[57]
			-	-	[67]
1,3,6-trihydroxy-2-methoxyxanthone **(194)**	C_14_H_10_O_6_	273	flowers	*S. nitens*	[57]
			flowers	*Syngonanthus* ssp.	[63]
1,3,6-trihydroxy-2,5-dimethoxyxanthone **(195)**	C_15_H_12_O_7_	303	flowers	*S. nitens*	[57]
			flowers	*Syngonanthus* ssp.	[63]
			Capitula, seeds	*E. buergerianum*	[67]
			Capitula, seeds	*E. cinereum*	[67]
			Capitula, seeds	*E. faberi*	[67]
3,4,8-trihydroxy-2,6-dimethoxyxanthone **(196)**	C_15_H_12_O_7_	303	flowers	*S. nitens*	[57]
1,3,6,8-tetrahydroxy-2,5-dimethoxyxanthone **(197)**	C_15_H_12_O_8_	319	flowers	*S. nitens*	[57]
			flowers	*Syngonanthus* ssp.	[63]
1,3,6,8-tetrahydroxy-5-methoxyxanthone **(198)**	C_14_H_10_O_7_	289	flowers	*Syngonanthus* ssp.	[63]
1,5,7-trihydroxy-3,6-dimethoxyxanthone **(199)**	C_15_H_12_O_7_	303	Capitula	*S. nitens*	[33]
			flowers	*Syngonanthus* ssp.	[63]
1,3,6-trihydroxy-5,7-dimethoxyxanthone **(200)**	C_15_H_12_O_7_	303	Capitula, seeds	*E. cinereum*	[67]
			Capitula, seeds	*E. faberi*	[67]
1,3,6-trihydroxy-2,7,8-trimethoxyxanthone **(201)**	C_16_H_14_O_8_	333	vines	*E. buergerianum*	[54]

MF = molecular formula.

#### 2.1.5. Saponins

Saponins are bioorganic compounds with at least one glycosidic linkage at C-3. The aglycone portion, called genin, sapogenin, or aglycone, may be composed of triterpenoid, steroid, or alkaloid. Triterpenoid saponin kind is the most widely distributed skeleton in the plant kingdom. The glycoside moiety may be composed of both hexoses and pentoses with different conformations. The lipophilic and hydrophilic composition of saponins confer a surfactant action, making saponins a class of metabolites with economic importance. The literature describes saponins’ biologic activities such as antibacterial, antitumoral, antifungal, anti-inflammatory, antiviral, insecticidal, etc. [98].

Zanatta et al. (2018) conducted the first study on the chemical composition and biological activity of the aerial parts of *Actinocephalus divaricatus* [79]. To make a fingerprint of methanolic extract, the authors chose to use a dereplication approach based on Liquid Chromatography coupled to High-Resolution Mass Spectrometry (HPLC-ESI-HRMSMS) as an analytical tool for the screening and structural determination.

Eight saponins were identified based on the fragmentation pattern and compared to literature data. The saponins **202–209** were identified in *Actinocephalus divaricatus* scapes and **204–208** in the leaves. A known nortriterpenoid saponin 3-*O*-β-D-glucuronopyranosyl-30-norolean-12,20(29)-dien-28-*O*-β-D-glucopyranosyl ester (**205**) was isolated and had its structure determined by MS/MS and NMR experiments and the data were compared with those published in the literature. Besides the mentioned compounds, others were identified but they could not be drawn.



Table 5 shows the names of saponins, the part of the plant where they were isolated, species, and authors.

**Table 5 molecules-27-07186-t005:** Saponins isolated from Eriocaulaceae species.

Compounds	MF	[M−H]^−^	Organ	Species	Reference
3-*O*-glucuronide oleanolic acid-28-*O*-hexose **(202)**	C_42_H_66_O_14_	793	Scapes	*Actinocephalus divaricatus*	[79]
3-*O*-glucuronide-29-hydroxyoleanolic acid-28-*O*-hexose esther **(203)**	C_42_H_66_0_15_	809	Scapes	*A. divaricatus*	[79]
3-*O*-[pentose-(1→3)-*O*-glucuronide]-30-norolean-12,20(29)-dien-28-*O*-hexose ester **(204)**	C_46_H_70_O_18_	909	Scapes, leaves	*A. divaricatus*	[79]
3-*O*-glucuronopyranosyl-30-norolean-12,20(29)-dien-28-*O*-glucopyranosyl ester **(205)**	C_41_H_62_0_15_	793	Scapes, leaves	*A. divaricatus*	[79]
3-*O*-[pentose-(1→4)-glucuronide] oleanolic acid-28-*O*-hexose **(206)**	C_47_H_74_O_18_	925	Scapes, leaves	*A. divaricatus*	[79]
3-*O*-glucuronide oleanolic acid **(207)**	C_36_H_56_O_9_	631	Scapes, leaves	*A. divaricatus*	[79]
3-*O*-pentose-(1→3)-*O*-glucuronide-30-norolean-12,20(29)-dien-28-oic acid **(208)**	C_40_H_60_O_13_	747	Scapes, leaves	*A. divaricatus*	[79]
30-norolean-12,20(29)-dien-28-*O*-hexose ester **(209)**	C_35_H_54_O_8_	601	Scapes	*A. divaricatus*	[79]

MF = molecular formula.

#### 2.1.6. Steroids

Steroids are complex lipophilic four-ringed organic molecules that act in the body to regulate cellular, tissue, and organ functions [99].

In 2008, Song et al. [100] described for the first time a phytochemical study on *Eriocaulon sieboldianum* (Eriocaulaceae), an aquatic annual herb of Korea, Japan, China, and Africa. Three stigmastane-skeleton sterols were isolated from the ethyl acetate soluble fraction of whole plant and the chemical structures were determined by spectroscopic and spectrometric techniques as IR, MS, and NMR. The isolated compounds were identified as one new compound, stigmasta-7,22-dien-3β,4β-diol (**210**), and two known compounds, stigmasterol 3-*O*-β-D-glucopyranoside (**211**) and stigmasta-5-en-3β-ol (β -sitosterol, **212**).



Table 6 shows the names of steroids, the part of the plant where they were isolated, species, and authors.

**Table 6 molecules-27-07186-t006:** Steroids isolated from Eriocaulaceae species.

Compounds	MF	[M−H]^−^	Organ	Species	Reference
Stigmasta-7,22-dien-3β,4β-diol **(210)**	C_29_H_48_O_2_	427	whole plant	*Eriocaulon sieboldianum*	[100]
stigmasterol 3-*O*-β-D-glucopyranoside **(211)**	C_35_H_58_O_6_	573	whole plant	*E. sieboldianum*	[100]
β-sitosterol **(212)**	C_29_H_50_O	413	whole plant	*E. sieboldianum*	[100]

MF = molecular formula.

#### 2.1.7. Quinones

Quinones are a large class of natural organic compounds and may be divided into three main groups: benzoquinone, naphthoquinone, and anthraquinone, with this last group as the most common cycle. Currently, more than 2000 quinones are known in nature, found in fungi, lichens, gymnosperms, and angiosperms. In families such as Rubiaceae, Fabaceae, and Boraginaceae, isolating quinones is very common [25], but according to the literature, only two quinones were isolated in Eriocaulaceae.

Kitagawa et al. (2004) [101] isolated for the first time the naphthoquinone 5-methoxy-3,4-dehydroxanthomegnin (**213**) from *Paepalanthus latipes* capitula methylene chloride extract. The cytotoxic evaluation showed a significant cytotoxic index of 35.8 mg mL^−1^ when compared with cisplatin (IC_50_ 41.9 mg mL^−1^), a cytotoxic substance used in antineoplasic therapy. Kitagawa carried out other biological assessment tests over the years, such as antitumor, immunomodulatory [102], antioxidant, anti-*helicobacter pylori* activity [103], and mutagenic [104].

In 2008, Fang and co-workers isolated, from an ethyl acetate fraction of the whole *Eriocaulon buergerianum*, the anthraquinone emodin (**214**). An antibacterial assay performed with the standard *Staphylococcus aureus* strain (ATCC 25923) showed a MIC 32 µg mL^−1^ [53].

Qiao et al. (2012) identified quinone **215** in a Chinese herbal medicine Gu-Jing-Cao (*Eriocaulon buergerianum*) by analyzing *m/z* obtained HPLC-DAD-ESI-MS^n^ [67].



Table 7 shows the names of anthraquinone, the part of the plant where they were isolated, species, and authors.

**Table 7 molecules-27-07186-t007:** Quinones isolated from Eriocaulaceae species.

Compounds	MF	[M−H]^−^	Organ	Species	Reference
5-methoxy-3,4-Dehydroxanthomegnin **(213)**	C_16_H_12_O_7_	315	Capitula	*Paepalanthus latipes*	[101]
Emodin **(214)**	C_15_H_10_O_5_	269	whole plant	*Eriocaulon buergerianum*	[53]
(R)-Semixanthomegnin **(215)**	C_15_H_12_O_6_	287	Capitula/seeds	*E. buergerianum*	[67]

MF = molecular formula.

#### 2.1.8. Phenolic Acid Derivatives

Phenolic or phenolcarboxylic acids are one of the main classes of plant phenolic compounds. They are found in a variety of plants and are mainly divided into two sub-groups: hydroxybenzoic and hydroxycinnamic acid. Phenolic acids are known for diverse biological applications such as antidiabetic, anticancer, neuro protective, and food preservative and skin care products [105].

Studying the whole plant of *Eriocaulon buergerianum*, Fang et al. (2008) isolated from different sub-fractions of ethyl acetate extract the phenolics vanillic acid (**216**, 190.0 mg), ferulic acid (**217**, 20.0 mg), and protocatechuic acid (**218**, 140 mg). All were isolated with a silica gel column or polyamide chromatography and elucidated by spectroscopic techniques. Additionally, they were tested against the pathogenic bacteria *Staphylococcus aureus* (ATCC 25923). As a result, **218** showed a MIC of 32 µg mL^−1^, **222** of 64 µg mL^−1^, and **217** of 256 µg mL^−1^ [53].

In 2012, studying the same species, Qiao et al. identified **216**, **218**, and caffeic acid (**219**) from the 70% methanol extract by HPLC-DAD-ESI-MS^n^ [67].



One colorless and amorphous caffeic acid derivative, 3-di-*E*-caffeoylglycerol **220**, was isolated from the ethanolic extract of *Paepalanthus microphyllus* capitula. The structure of **220** was characterized by spectroscopic and spectrometry methods [46].



The methanolic extract from *Leiothrix flavescens* capitula was partitioned with EtOAc and submitted to separation by HSCCC. Afterwards, a fraction was chromatographed with Sephadex LH-20 to yield 1,3-*O*-diferuloilglicerol **221**. The structure of **221** was characterized by spectroscopic and spectrometry methods [106].



Table 8 shows the names of phenolic acids derivatives, the part of the plant where they were isolated, species, and authors.

**Table 8 molecules-27-07186-t008:** Phenolic acids derivatives isolated from Eriocaulaceae species.

Compounds	MF	[M−H]^−^	Organ	Species	Reference
Vanillic acid **(216)**	C_8_H_8_O_4_	167	whole plant	*Eriocaulon buergerianum*	[53]
			Capitula, seeds	*E. buergerianum*	[67]
Ferulic acid **(217)**	C_10_H_10_O_4_	193	whole plant	*E. buergerianum*	[53]
Protocatechuic acid **(218)**	C_7_H_6_O_4_	153	whole plant	*E. buergerianum*	[53]
			Capitula, seeds	*E. buergerianum*	[67]
Caffeic acid **(219)**	C_9_H_8_O_4_	180	Capitula, seeds	*E. buergerianum*	[67]
3-di-*E*-caffeoylglycerol **(220)**	C_21_H_20_O_9_	415	Capitula	*Paepalanthus microphyllus*	[67]
1,3-*O*-diferuloilglicerol **(221)**	C_23_H_24_O_9_	443	Capitula	*Leiothrix flavescens*	[67]

MF = molecular formula.

#### 2.1.9. Tocopherol

The EtOAc fraction of *Eriocaulon buergerianum* Koern. capitula was chromatographed in a silica gel column. A fraction after the mentioned procedure was separated by MPLC to yield γ-tocopheryl acetate **222**. The structure of **222** was established by spectral methods [44].



### 2.2. Ligand-Based Virtual Screening

The Random Forest algorithm (RF) models were generated following the five-fold cross validation procedure [107,108]. Table 9 shows the statistical performances for each model.

The performance characteristics of the nine models created revealed their predictive power and reliability, and the Receiver Operating Characteristic (ROC) curve and Matthews Correlation Coefficient (MCC) gave information on their performance and robustness.

The Eriocaulaceae dataset with a total of **222** compounds was reduced to 206 molecules (16 structures with undefined sugar units were removed for virtual screening analysis) and analyzed in each model.

In each model, we analyzed the applicability domain (APD) for each molecule to consider its prediction, that is, molecules within the APD of the model have reliable predictions and outside the APD have unreliable predictions. Table 10 shows a summary of how many molecules were inside the APD of each model, and how many of these molecules have an active potential prediction against parasites that cause neglected diseases.

Table 11 shows that for the model against *Trypanosoma cruzi* parasitic form Trypomastigote, no molecule that remained inside the APD showed active potential. In the *T. cruzi*—amastigote model only two molecules were predicted to be active out of the 158 molecules that were inside the APD and in the *Leishmania infantum*—amastigote model only eight molecules were predicted to be active out of the 200 that were inside the APD.

The *Leishmania infantum*—promastigote model had the highest active prediction potential, with molecules reaching a 96% probability of being active. Table 11 shows the five molecules with the highest active potential for the models created.

Analyzing the predictions of molecules isolated from Eriocaulaceae, we can observe that some molecules have multitarget potential, capable of being active for different parasites.

Molecules **194** and **196**, xanthones, were shown to have active potential probability against *Aedes aegypti* (larvicidal activity), *Schistosoma mansoni*, *Leishmania amazonensis* (promastigote), *Leishmania infantum* (promastigote), *Trypanosoma cruzi* (amastigote). Molecule **194** also has active potential against *L. amazonensis* (amastigote) and **196** against *T. cruzi* (epimastigote).

The xanthone **200** and the saponin **202** have very similar multitarget active potential probability against *L. amazonensis* (promastigote), *L. infantum* (amastigote and promastigote), and *Ae. aegypti* (larvicidal activity).

Among these four molecules identified with multitarget potential, the most promising one, which has the highest probability of active potential, is xanthone **194**. Table 12 shows the molecules with multitarget potential and their respective probabilities of active potential.

## 3. Materials and Methods

### 3.1. Database

A selection of data was made by a search in Scifinder^®^, ScienceDirect, Scielo, Web of Science^TM^, and Scopus^®^ databases by the *Portal de Periódicos CAPES* website. About 70 articles were found about isolated and/or identified compounds from 1969 to 2022. As a keyword were used “Eriocaulaceae”, “Eriocaulaceae compounds”, “Flavonoids of Eriocaulaceae”, “Naphtopyranones of Eriocaulaceae”, and the genus plus “compounds”.

### 3.2. Dataset

Five sets of chemical structures with known activity for *Schistosoma mansoni*, *Leishmania infantum*, *Leishmania amazonensis*, *Trypanosoma cruzi*, and *Aedes aegypti* were selected from the ChEMBL database [109,110,111], for the construction of the nine predictive model. Table 13 shows the information about each dataset.

The dataset built in this review with compounds isolated from Eriocaulaceae was used to predict the potential activity against neglected diseases.

For all structures, SMILES (Simplified Molecular Input Line Entry System) codes were used as input data for Marvin 18.10.0, 2018 (ChemAxon) [112] and Standardizer software (JChem 18.10.0, 2018; ChemAxon) [112] to convert the chemical structures into curated and canonical representations. This standardization is of paramount importance to create consistent compound libraries and is done by the following steps: addition of hydrogens, aromatization, generation of 3D structure, and exporting the compounds in SDF format. For a more detailed description on how the dataset was curated, please refer to the workflows described by Fourches et al. [113,114,115].

### 3.3. DRAGON 7.0 Descriptors

Molecular descriptors are used to calculate the physicochemical properties of the molecules from each molecule set. To obtain the molecular descriptors, the DRAGON 7.0 program [116] was used. The DRAGON 7.0 software can calculate 5270 molecular descriptors, covering various approaches. These molecular descriptors are arranged in 30 logical blocks [116].

### 3.4. Random Forest Model

The KNIME 4.4.0 software (KNIME version number 4.4.0, the Konstanz Information Miner Copyright, 2003–2021, Zurich, Switzerland) [117] was used to perform the analyses and to generate the *in silico* model. Datasets of molecules, along with their calculated descriptors and class variables, were imported from DRAGON 7.0 software. Each dataset was divided using the “partitioning” tool, with the “stratified sample” option, to create a training set and an external test set, which represented 80% and 20% of the compounds, respectively. Although the compounds were selected randomly, the same proportion of active and inactive samples was maintained in both sets.

For external validation, we employed five-fold cross-validation [107,108] using randomly selected, stratified groups, meaning that the entire data set was partitioned five times into a modeling set (training set) including 80% of the compounds in the set, and the external cross validation data set, comprising the remaining 20% of the compounds in the data set. After this, only the modeling set was used to build the models and then the models were validated with the external cross validation technique. Descriptors were selected and modeled following a five-fold external cross validation procedure using the RF [107,108]. A total of 100 parameters were selected for RF for all generated models, which is the number of trees constructed, and 1550953075932 seeds generated random numbers for the model.

By using KNIME nodes, the most important descriptors in generating each prediction model were evaluated. The external performances of the selected models were analyzed for sensitivity (true positive rate, i.e., active rate), specificity (true negative rate, i.e., inactive rate), and accuracy (overall predictability). The positive (PPV) and negative (NPV) predictive values inform us about the probability of predicted positives (PPV) and negatives (NPV) being the true positives and negatives, respectively. The sensitivity and specificity of the ROC curve were also found to describe true performance with more clarity than accuracy.

The model was also analyzed by the Matthews coefficient, a way to evaluate the model globally from the results obtained from the confusion matrix. The MCC is a correlation coefficient between observed and predictive binary classifications. It results in a value between −1 and +1, where a coefficient of +1 represents a perfect forecast, 0 is a random forecast, and −1 indicates total disagreement between forecast and observation [118].

The MCC can be calculated from the following formula:(1)$$MCC=\frac{VP\times VN-FP\times FN}{\surd \left(VP+FP\right)\left(VP+FN\right)\left(VN+FP\right)\left(VN+FN\right)}$$
where *VP* is the true positive value, *VN* is the true negative value, *FP* is the value of false positives, and *FN* is the value of false negatives.

The APD was used to analyze the compounds of the test sets to evaluate whether their predictions were reliable. The APD is based on Euclidean distances, and similarity measures between the descriptors of the training set are used to define the applicability domain. Thus, if a test set compound has distances and similarity beyond this limit, its prediction is unreliable.

The APD calculation is performed by using the formula:APD = *d* + *Zσ*(2)
where *d* and σ are the Euclidean distances and the standard mean deviation, respectively, of the compounds in the training set. *Z* is an empirical cut-off value. In this work, the *Z* value used was 0.5 [119,120].

## 4. Conclusions

Eriocaulaceae, a pantropical family, comprises around 1200 species divided in 10 genera. *Eriocaulon*, its largest genus, comprising about 478 species, has 13 species phytochemically studied and *Paepalanthus*, the second largest genus, comprising about 400 species, has 26 studied species. As far we know, 130 flavonoids were isolated in 57 species divided into *Eriocaulon*, *Paepalanthus*, *Syngonanthus*, and *Leiothrix* genera; 36 naphthopyranones were isolated in 27 species from *Eriocaulon*, *Paepalanthus*, and *Actinocephalus*; 21 compounds were derivative from *Paepalanthus* fungi; 14 xanthones were isolated in six species from *Eriocaulon*, *Syngonanthus*, and *Leiothrix*; eight saponins from *Actinocephalys divaricatus*; three steroids from *Eriocaulon sieboldianum*; three quinones from *Eriocaulon buergerianum* and *Paepalanthus latipes*; six phenolic acid derivatives from *Eriocaulon buergerianum*, *Paepalanthus microphyllus,* and *Leiothrix flavescens*, and finally, one tocopherol derivative from *Eriocaulon buergerianum*.

Many of these compounds had their in vitro biological activities tested, and their mutagenic, antimicrobial, antioxidant, anti-inflammatory, antitumor, and other activities were confirmed.

The computational study showed that the molecules isolated from the Eriocaulaceae family have active potential against neglected diseases: leishmaniasis, schistosomiasis, Chagas disease, and dengue fever. Finding promising molecules with multitarget potential, which were xanthones **194**, **196,** and **200** and saponin **202**, was also possible, with xanthone **194** as the most promising of them.

These data present the relevance of this review, which shows the structural variation of the compounds isolated in the family and in vitro and in vivo biological activities carried out so far and highlights the possibility of new compounds being known, given the number of species that have not been chemically studied. In addition, the chemical and biological knowledge of a species can contribute to the preservation of the species, an important fact, since many Eriocaulaceae species are at risk of extinction.

## Figures and Tables

**Table 9 molecules-27-07186-t009:** Summary of the parameters of the results obtained in the RF models.

Models	Parasitic Stage	Sensitivity	Specificity	Accuracy	PPV	NPV	MCC	ROC Curve Area under the Curve
*Aedes aegypti*	Larva	0.95	0.75	0.85	0.82	0.92	0.72	0.97
*Leishmania amazonensis*	Amastigote	0.6	0.65	0.63	0.6	0.65	0.35	0.71
	Promastigote	0.7	0.88	0.81	0.66	0.8	0.61	0.89
*Leishmania infantum*	Amastigote	0.81	0.9	0.85	0.81	0.9	0.71	0.94
	Promastigote	0.77	0.78	0.78	0.75	0.8	0.56	0.89
*Trypanosoma cruzi*	Trypomastigote	0.79	0.83	0.81	0.83	0.8	0.62	0.86
	Amastigote	0.81	0.78	0.79	0.79	0.79	0.59	0.89
	Epimastigote	0.78	0.70	0.72	0.76	0.70	0.44	0.82
*Schistosoma mansoni*	Adult	0.96	0.86	0.91	0.83	0.97	0.81	0.98

PPV = Positive Predictive Values; NPV = Negative Predictive Values; MCC = Matthews Correlation Coefficient; ROC = Receiver Operating Characteristic.

**Table 10 molecules-27-07186-t010:** Prediction information.

Models	Parasitic Stage	Applicability Domain	Active Prediction	Probability to Be Active
*Aedes aegypti*	Larva	166	23	51–87%
*Leishmania amazonensis*	Amastigote	131	57	52–81%
	Promastigote	140	81	51–88%
*Leishmania infantum*	Amastigote	200	8	51–66%
	Promastigote	148	122	51–96%
*Trypanosoma cruzi*	Trypomastigote	183	-	-
	Amastigote	158	2	54%, 68%
	Epimastigote	162	99	51–74%
*Schistosoma mansoni*	Adult	96	71	51–87%

**Table 11 molecules-27-07186-t011:** Five molecules with the highest active potential probability of each model.

Models	Parasitic Stage	Molecules	Probability to Be Active
*Aedes aegypti*	Larva	**206** **196** **158** **194** **205**	87% 82% 80% 78% 63%
*Leishmania amazonensis*	Amastigote	**25** **195** **30** **24** **52**	81% 81% 80% 80% 79%
	Promastigote	**196** **195** **194** **42** **59**	88% 84% 83% 75% 75%
*Leishmania infantum*	Amastigote	**200** **153** **202** **34** **63**	66% 58% 57% 54% 53%
	Promastigote	**202** **124** **123** **200** **125**	96% 92% 92% 91% 91%
*Trypanosoma cruzi*	Trypomastigote	-	-
	Amastigote	**194** **196**	69% 54%
	Epimastigote	**204** **182** **166** **44** **177**	74% 70% 69% 68% 68%
*Schistosoma mansoni*	Adult	**122** **205** **143** **128** **145**	87% 83% 82% 82% 81%

**Table 12 molecules-27-07186-t012:** Molecules with multitarget potential and their respective active potential probabilities.

Molecule	Parasitic Stage	Probability to Be Active
**194**	*Leishmania amazonensis* (amastigote) *Leishmania amazonensis* (promastigote) *Leishmania infantum* (promastigote) *Trypanosoma cruzi* (amastigote) *Schistosoma mansoni* *Aedes aegypti* (larva)	77% 83% 81% 69% 62% 78%
**196**	*L. amazonensis* (promastigote) *L. infantum* (promastigote) *T. cruzi* (amastigote) *T. cruzi* (epimastigote) *S. mansoni* *Ae. aegypti* (larva)	88% 81% 55% 54% 55% 82%
**200**	*L. amazonensis* (amastigote) *L. infantum* (amastigote) *L. infantum* (promastigote) *Ae. Aegypti*	59% 57% 91% 54%
**202**	*L. amazonensis* (amastigote) *L. infantum* (amastigote) *L. infantum* (promastigote) *Ae. Aegypti*	64% 57% 96% 57%

**Table 13 molecules-27-07186-t013:** ChEMBL databases.

Database	Parasitic Stage	Total Molecules	Active	Inactive	ChEMBL ID
*Aedes aegypti*	Larva	173	pIC_50_ ≥ 4.15	pIC_50_ < 4.15	CHEMBL613468
*Leishmania amazonensis*	Amastigote	226	pIC_50_ ≥ 5	pIC_50_ < 5	CHEMBL612877
	Promastigote	576	pIC_50_ ≥ 4.8	pIC_50_ < 4.8	
*Leishmania infantum*	Amastigote	372	pIC_50_ ≥ 5.5	pIC_50_ < 5.5	CHEMBL612848
	Promastigote	388	pIC_50_ ≥ 4.7	pIC_50_ < 4.7	
*Trypanosoma cruzi*	Trypomastigote	1694	pIC_50_ ≥ 5	pIC_50_ < 5	CHEMBL368
	Amastigote	1157	pIC_50_ ≥ 4.71	pIC_50_ < 471	
	Epimastigote	1884	pIC_50_ ≥ 4.7	pIC_50_ < 4.7	
*Schistosoma mansoni*	Adult	360	pIC_50_ ≥ 6	pIC_50_ < 6	CHEMBL3563

## Data Availability

Not applicable.

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
