# Peer review of "Review on Compounds Isolated from Eriocaulaceae Family and Evaluation of Biological Activities by Machine Learning"

_molecules, 2022, doi:10.3390/molecules27217186_

Round 1
Reviewer 1 Report
Language is very weak throughout and article is very long. Kindly decrease the length of it. Some structures are blurred.
Author Response
1) The language has been corrected by the translation company TIKINET. The certificate is
attached.
2) Due to the size of the tables and chemical structures, the article was long, but in the
written part a summary of the procedures for isolation and/or identification of
compounds was performed. Biological activities were also presented when tested. We
kindly ask you to consider the article at its current size.
3) The blurred structures have been rectified.

Reviewer 2 Report
Eriocaulaceae plants are not very important pharmaceutically. This is a thorough compilation of compounds isolated from the family of Eriocaulaceae.
The structure of compounds in the text needs to be in same size. The authors do not know how to print the scientific name of organisms. In a chapter or a table/figure, the genus name of an organism needs to be spell fully, at least when it first apeared. The first letter of the species adding name of an organism needs to be in small form always. Correct the scientific names of organisms in the text, tables and references
Author Response
1) The figures size was corrected.
2) The scientific name of genera was completely written when appeared for the first time
in each paragraph. Also, the scientific names of species were corrected in the references.
3) The second name of a species are all in lower cap now.

Reviewer 3 Report
The authors have carried out a comprehensive review of the literature on compounds isolated from Eriocaulaceae plant materials and some associated fungi, summarised their isolation conditions, structures, and where available identified their bioactive properties. Two hundred and twenty-two different compounds from sixty-eight species were noted. These comprised mainly of flavonoids and naphthopyranones, but also included xanthones, saponins, steroids, quinones, phenolics, tocopherol and fungal metabolites. Based on known structures of two hundred and six of the compounds, ligand-based virtual screening has identified for the first time three xanthones and one saponin as having potentially potent anti-parasitic properties and meriting further detailed study.
This is a comprehensive overview of data from a very diverse set of reported studies and well summarises the potential of bioactive factors from Eriocaulaceae for treatment of diseases or in promotion of health. Follow-on analysis has identified four compounds with good potential to modulate or treat intractable parasitic infections.
The review will provide an excellent reference for workers in the field but does require an English check. For example, ln 129 'draw' should be 'drawn', and ln 372 'partionated' should be 'partitioned'.
Author Response
1) The language has been corrected by the translation company TIKINET. The certificate is
attached.
2) The word “draw” in line 129 and 'partionated' now in line 374 were corrected to
“drawn” and “partitioned”, respectively.

Reviewer 4 Report
The presented study is a review of the chemical composition and biological activity of compounds isolated from plants of the Eriocaulaceae family. The review is written very precisely, for this purpose data from a considerable amount of literature was processed. I also appreciate the respectable scope of the entire manuscript without significantly affecting the clarity and expressiveness of the text.
I recommend the publication of this manuscript in the form presented and rate it as beneficial.
Author Response
1) None correction was required.

Round 2
Reviewer 1 Report
-